# eIF4E S209 phosphorylation licenses myc- and stress-driven oncogenesis

Hang Ruan[1,2†], Xiangyun Li[1,2,3†], Xiang Xu[3,4], Brian J Leibowitz[1,2], Jingshan Tong[2,5], Lujia Chen[2,6], Luoquan Ao[3,4], Wei Xing[3,4], Jianhua Luo[1], Yanping Yu[1], Robert E Schoen[7], Nahum Sonenberg[8], Xinghua Lu[2,6], Lin Zhang[2,5], Jian Yu[1,2]*

[1]Department of Pathology, University of Pittsburgh School of Medicine, Pittsburgh, United States; [2]UPMC Hillman Cancer Center, Pittsburgh, United States; [3]Department of Stem cell and regenerative medicine, Daping Hospital, Army Medical University, Chongqing, China; [4]Central laboratory, State key laboratory of trauma, burn and combined Injury, Daping Hospital, Chongqing, China; [5]Department of Pharmacology and Chemical Biology, University of Pittsburgh School of Medicine, Pittsburgh, United States; [6]Department of Biomedical informatics, University of Pittsburgh School of Medicine, Pittsburgh, United States; [7]Departments of Medicine and Epidemiology, University of Pittsburgh, Pittsburgh, United States; [8]Department of Biochemistry, Goodman Cancer Research Centre, McGill University, Montreal, Canada

*For correspondence:
yuj2@upmc.edu

[†]These authors contributed equally to this work

**Abstract** To better understand a role of eIF4E S209 in oncogenic translation, we generated *EIF4E*[S209A/+] heterozygous knockin (4EKI) HCT 116 human colorectal cancer (CRC) cells. 4EKI had little impact on total eIF4E levels, cap binding or global translation, but markedly reduced HCT 116 cell growth in spheroids and mice, and CRC organoid growth. 4EKI strongly inhibited Myc and ATF4 translation, the integrated stress response (ISR)-dependent glutamine metabolic signature, AKT activation and proliferation in vivo. 4EKI inhibited polyposis in *Apc*[Min/+] mice by suppressing Myc protein and AKT activation. Furthermore, p-eIF4E was highly elevated in CRC precursor lesions in mouse and human. p-eIF4E cooperated with mutant *KRAS* to promote Myc and ISR-dependent glutamine addiction in various CRC cell lines, characterized by increased cell death, transcriptomic heterogeneity and immune suppression upon deprivation. These findings demonstrate a critical role of eIF4E S209-dependent translation in Myc and stress-driven oncogenesis and as a potential therapeutic vulnerability.

## Introduction

Colorectal cancer (CRC) is a leading cause of cancer-related death worldwide (*Siegel et al., 2019*). The gatekeeper tumor suppressor *APC* is mutated in 85% of CRCs and leads to increased Wnt/Myc signaling which cooperates with mutational activation of RAS/RAF/ERK (50–80%) and PI3K/AKT/mTOR pathways (10–15%) to promote CRC initiation and progression (*Vogelstein et al., 2013*). Emerging evidence suggests that oncogenic drivers such as Myc do not simply increase 'physiologic' proliferation (*Dang, 2016*), but engender 'oncogenic' growth and hallmarks such as altered metabolism, resistance to cell death, metastasis, and immune evasion (*Hanahan and Weinberg, 2011*). Since direct targeting Myc (*Dang et al., 2017*) or mutant *KRAS* (*Vogelstein et al., 2013*) has not been successful in the clinic, intense interest remains to identify potential druggable targets in their regulatory circuitry.

mRNA translation is a highly energy-consuming and regulated process, and a converging target of oncogenic drivers (*Pelletier et al., 2015*; *Truitt and Ruggero, 2016*). The assembly of cap-binding complex eIF4F, consisting of the eukaryotic translation initiation factor 4E (eIF4E), RNA helicase eIF4A and scaffold eIF4G, is the rate-limiting step in translation initiation, which entails the unwinding of the secondary structure in the mRNA 5′UTR to facilitate recruitment of the 43S pre-initiation complex (PIC) containing the 40S ribosome and the eIF2α-GTP-Met-tRNA ternary complex for AUG codon recognition. Phosphorylation of eIF4E (S209) (p-4E, thereafter) and its inhibitor 4E-BP1 (i.e. T37/T46, S65/T70) is elevated in a variety of cancers due to activated RAS/RAF/ERK and PI3K/AKT/mTOR signaling (*Martineau et al., 2013*). Map kinase-interacting kinase 1 and 2 (Mnk1 and Mnk2) are activated by ERK or p38 MAPKs in response to a variety of extracellular stimuli to phosphorylate eIF4E (*Wang et al., 1998*). Constitutive or inducible p-4E is mediated by Mnk1/2 that are dispensable for normal development (*Ueda et al., 2004*; *Ueda et al., 2010*). p-4E is required for cellular transformation (*Topisirovic et al., 2004*) but dispensable for normal development in mice (*Furic et al., 2010*). 4E-BP1 and 4E-BP2 in their un- or hypo-phosphorylated forms are believed to inhibit eIF4E-eIF4G binding and even p-4E levels (*Martineau et al., 2013*). Genetic ablation of either or both *Eif4ebp1* and *Eif4ebp2* in mice leads to metabolic defects not spontaneous tumorigenesis (*Le Bacquer et al., 2007*). Genetic alterations in *EIF4E* or *EIF4E4EBP1* and 2 are extremely rare or absent in human cancer. The oncogenic function of p-4E and its regulation remain to be better defined and likely go far beyond increased cap binding or global mRNA translation (*Martineau et al., 2013*).

Phosphorylation of eIF2α (S51, p-eIF2α, thereafter) is the core of evolutionarily conserved 'integrated stress response' (ISR) (*Hetz et al., 2013*; *Tabas and Ron, 2011*; *Tameire et al., 2015*; *Cubillos-Ruiz et al., 2017*) and elevated in many cancers including CRC (*Schmidt et al., 2019*; *Schmidt et al., 2020*). In mammals, four eIF2α kinases are activated by distinct stresses such as nutrient deficiency, misfolded proteins, viral infection, or oxidative stress, and GCN2 is activated by amino acid starvation and conserved from yeast to human (*Castilho et al., 2014*). Elevated p-eIF2α inhibits global cap-dependent translation, while facilitates translation of stress-related proteins such as ATF4 and CHOP to regulate adaptation and recovery through widespread changes in transcription, translation, and metabolism. Failure to adapt leads to unresolved ISR, persistent CHOP elevation, and apoptosis via the induction of DR5 and BH3-only proteins (*Tabas and Ron, 2011*; *Harding et al., 2003*).

In the current study, we sought to better understand the oncogenic role of p-4E. Using *EIF4E*$^{S209A/+}$ knockin (4EKI) human colon cancer cells and mice, human CRC organoids and adenoma samples, we uncovered a key role of p-4E in Myc and ATF4 translation, which promotes ISR-dependent glutamine metabolism, AKT signaling and oncogenic proliferation. Mutant *KRAS* cooperated with Myc to promote p-4E and ISR-dependent glutamine addiction and transcriptomic heterogeneity upon deprivation. Our findings support a critical role of eIF4E S209-dependent translation in Myc- and stress- driven CRC initiation and progression.

## Results

### eIF4E S209 regulates colon cancer cell growth and Myc translation

p-4E and p-4E-BP1 (S65/T70) are highly elevated in many types of cancers including CRC (*Truitt and Ruggero, 2016*; *Martineau et al., 2013*; *Fan et al., 2009*), while their mRNA or total protein levels do not increase significantly in the Cancer Genome Atlas (TCGA) CRC cohort (n = 640) (*Figure 1—figure supplement 1A*). To understand the oncogenic function of p-4E, we created *EIF4E*$^{S209A/+}$ KI (4EKI thereafter) HCT 116 colon cancer cells using AAV-mediated gene editing (*Figure 1A–B*, *Figure 1—figure supplement 1B–C*). Compared to HCT 116 parental (WT) cells under normal growth conditions, 4EKI cells showed a strong reduction in p-4E and unexpectedly in p-4E-BP1 (S65/T70), p-AKT(S473), with no change in total eIF4E, 4E-BP1, p-4E-BP1(T37/46), p-mTOR (2448), p-S6 or p-ERK (*Figure 1A*). 4EKI slightly reduced 2D growth after day 5, as well as steady-state ATP levels (by 40%) and proliferation (by 20%) (*Figure 1C*, *Figure 1—figure supplement 1D–F*). 4EKI modestly reduced clonogenic and anchorage-independent growth, and severely impaired 3D spheroid growth (by 80%) (*Figure 1D–F*). We were unable to expand or obtain any homozygous 4EKI/KI HCT 116 clone, or heterozygous 4EKI clone in RKO or HT29 cells despite repeated attempts.

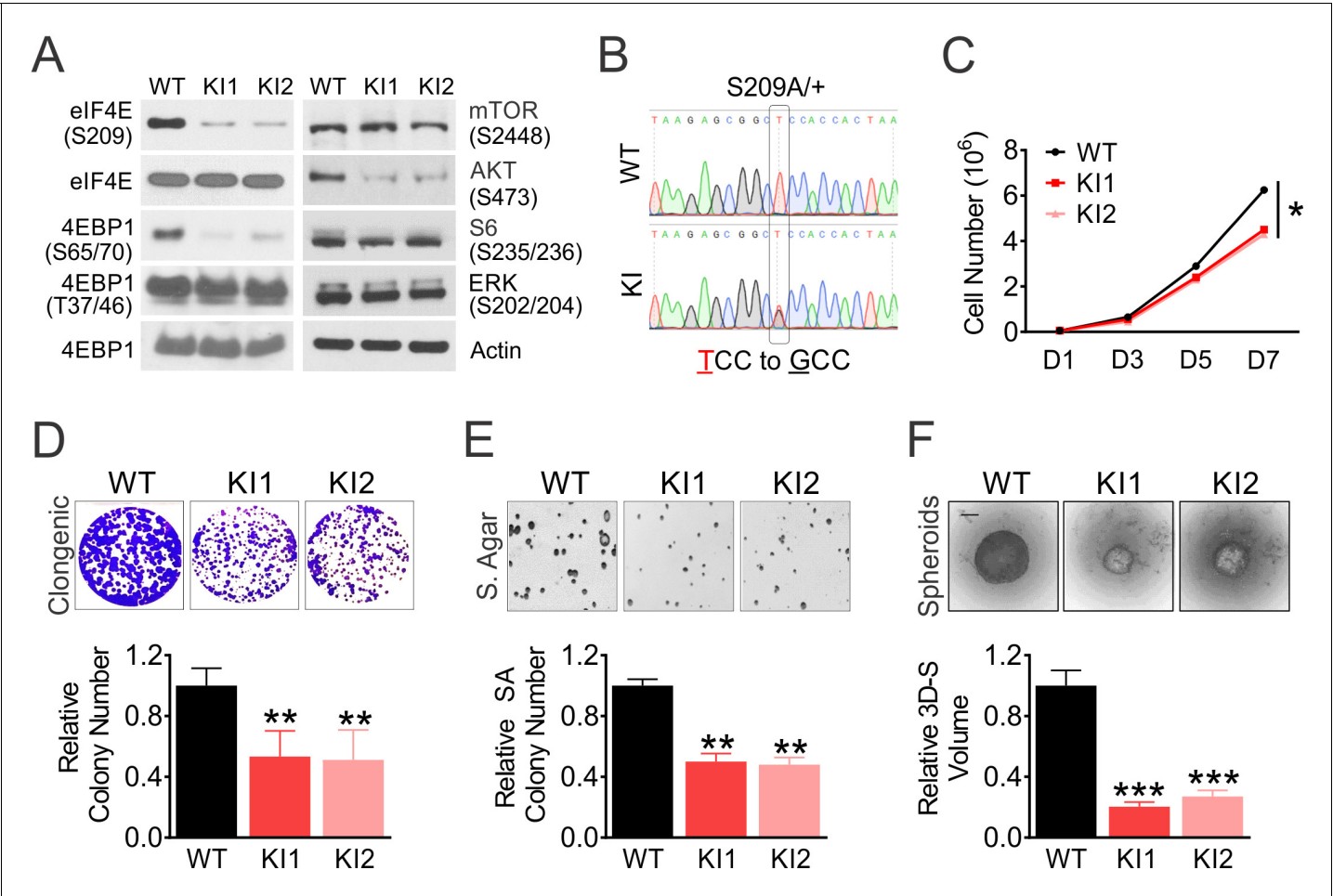

**Figure 1.** eIF4E S209 regulates CRC cell growth. Isogenic HCT116 WT and two independent *EIF4E^{S209A}* knockin (S209A/+, 4EKI1, and 4EKI2) clones were analyzed for growth. (**A**) The indicated proteins in log-phase cells were analyzed by western blotting. Actin is the loading control. (**B**) Genomic DNA sequencing confirmed T to G (S209A) change in 4EKI cells. (**C**) Cell growth monitored for 7 days by counting. (**D–F**) Representative images and quantification (bottom) of cell growth in clonogenic assay, soft agar, and spheroids on day 10, day 14, and day 7, respectively. (**F**) Scale bar: 10 μM. WT values were set at 100%. C, D, E, F, values are mean+s.d. (n = 3). *p<0.05, **p<0.01, ***p<0.001 (Student's t-test, two tailed). WT vs. KI.

The online version of this article includes the following figure supplement(s) for figure 1:

**Figure supplement 1.** Generation and characterization of *EIF4E^{S209A/+}* KI HCT 116 cells.

We next examined the effects of 4EKI on translation in unstressed HCT 116 cells. 4EKI slightly reduced global mRNA translation, as measured by polysome profiling and activities of cap-dependent Luciferase and GFP reporters (by ~20%) (*Figure 2A–C*, *Figure 2—figure supplement 1*). eIF4G and 4E-BP1 bind to eIF4E competitively. In untransformed cells such as mouse fibroblast cells (MEFs), serum or growth factor stimulates cap (m7GTP) binding of eIF4G with 4E-BP1 dissociation (*Martineau et al., 2013*). In HCT 116 cells, serum enhanced eIF4G binding without 4E-BP1 dissociation. 4EKI reduced serum-induced eIF4G binding and increased 4EBP1 dissociation. eIF4E-cap binding was not affected by either serum or 4EKI (*Figure 2D*). We then analyzed several well-established eIF4E targets (*Pelletier et al., 2015*), and found a strong reduction of Myc and MMP7 protein in 4EKI cells, correlated with preferential reduction in the polysomal over total cellular mRNAs. Other known 4E targets (Cyclin D, Bcl-xL) or 4E itself, a Myc target, showed little difference in either protein or mRNA (*Figure 2E–H*). eFT508 is a highly selective Mnk1/2 inhibitor and reduces p-4E (*Xu et al., 2019*). eFT508 treatment impaired the growth of three CRC organoids at 48 hr, and abrogated p-4E and reduced levels of p-4EBP1(S65/T70) and Myc at 24 hr, without affecting total eIF4E (*Figure 2I–J*). These data demonstrate that p-4E promotes optimal growth of CRC cells by maintaining p-4E-BP1, AKT signaling and translation of oncogenic targets such as Myc and MMP7.

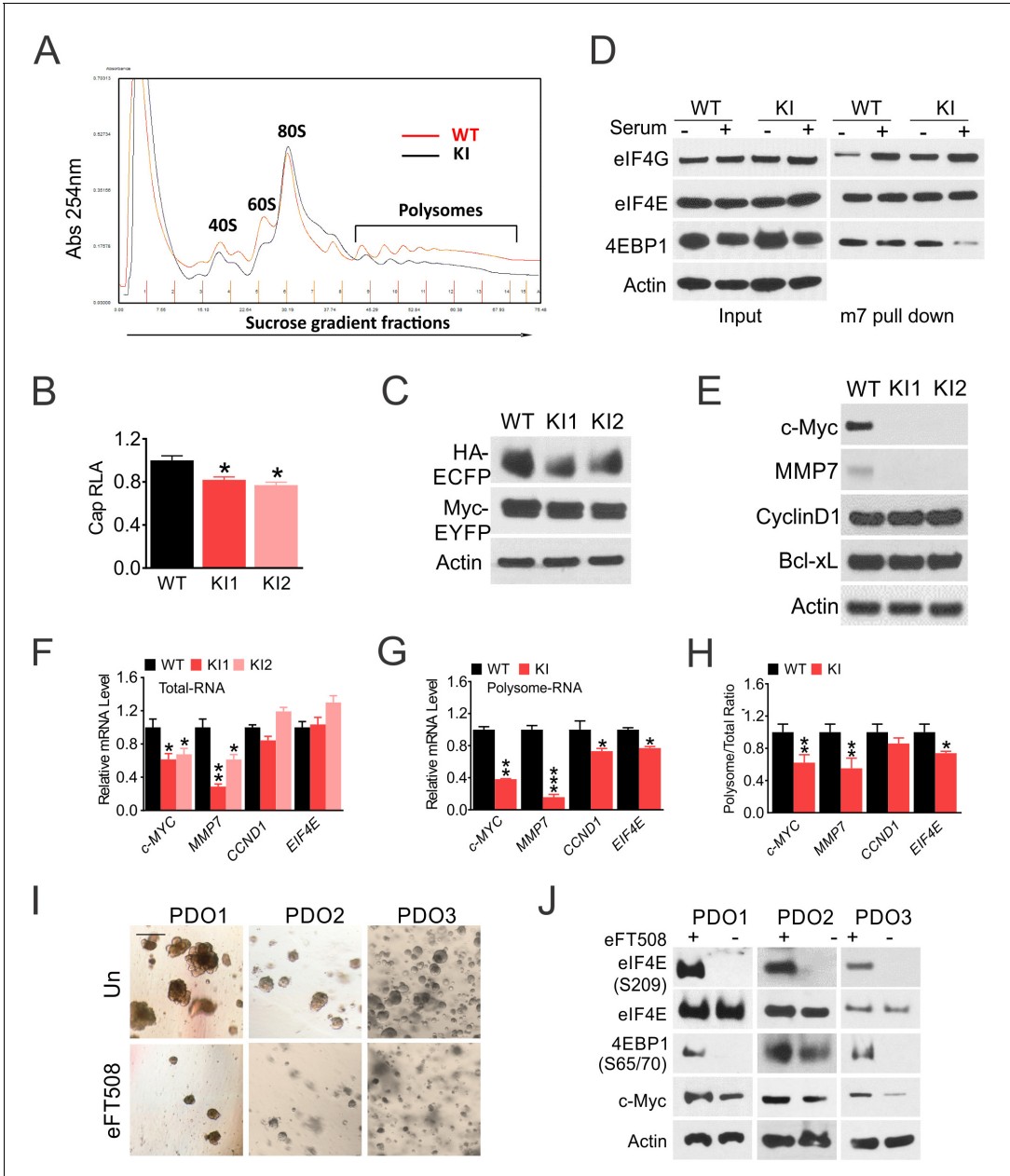

**Figure 2.** eIF4E S209 regulates Myc translation in CRC. WT and 4EKI cells were analyzed for translation. (**A**) Polysome profiling of log-phase cells. (**B**) Cap-dependent luciferase reporter activities were measured 24 hr after transfection with pcDNA-LUC reporter. (**C**) Translation of a bi-cistronic construct pYIC encoding cap-dependent and -independent reporters was analyzed by western blotting 24 hr after transfection. (**D**) The cells were subjected to serum starvation for 48 hr then stimulated with 10% FBS (+) for 4 hr. Binding of indicated eIF4F components (E, G, and 4E-BP1) to a synthetic cap analog ($m^7$GTP) or their levels in the total lysate (input) were analyzed by western blotting. (**E**) The indicated eIF4E targets were analyzed by western blotting. Actin is the loading control. (**F**) The indicated transcripts in total cellular and (**G**) in polysomal RNAs were analyzed by RT-PCR. (**H**) The ratio of mRNA in polysome and total RNA normalized to WT (100%). N = 3. (**B, F-H**), values are mean+s.d. (n = 3). *p<0.05, **p<0.01 (Student's t-test, two tailed). WT vs. KI. (**I**) Representative images of three CRC organoids treated with eFT508 (10 μM) for 48 hr. Scale bar: 100 μM. (**J**) The indicated proteins in CRC organoids treated with eFT508 at 24 hr were analyzed by western blotting.

The online version of this article includes the following figure supplement(s) for figure 2:

**Figure supplement 1.** Schematics of pCMV-Luc reporter and pYIC reporter encoding cap-dependent translation of Myc-tagged YFP and IRES-dependent HA-tagged CFP.

eIF4E S209 promotes colon cancer growth in vivo through Myc and the ISR. c-Myc is the critical CRC driver and essential for cell proliferation, and controls transcription of up to 14% of the genome and virtually every aspect of metabolism (*Dang, 2016*). Myc regulates gene expression via target-specific and more general and target-independent mechanisms (*Baluapuri et al., 2020*). This presents a challenge and critical need to uncover p-4E-dependent Myc transcriptional programs in CRC. We compared transcriptomes in WT and 4EKI cells using cDNA microarrays (*Figure 3—figure supplement 1A*). Significantly downregulated genes in 4EKI cells (89, 3-fold or more) were mapped to the ISR, including upstream regulators (ATF4, CHOP, ATF3, GADD34/45B), and effectors in translation (tRNA synthetases), amino acid and glutamine (Gln) metabolism, and apoptosis (*Figure 3A*, *Figure 3—figure supplement 1B*, *Supplementary file 1A*). Significantly upregulated genes in 4EKI cells (70, 3-fold or more) were mapped to DNA replication and collagen formation, in addition to stress (*Figure 3—figure supplement 1C–D*). We detected mRNAs of *ATF4*, *DDIT3* (encoding CHOP) and several Gln metabolic genes (*SLC1A5* (ASCT2), *ASNS, GLS, GOT1, SLC7A5*) in total cellular and polysomal RNAs (*Figure 3B–C*, *Figure 3—figure supplement 1E–F*). Reduction of *ATF4* and *DDIT3* in 4EKI cells was more significant in polysomal RNAs, compared to ISR effectors. SiRNA of *MYC, ATF4,* or *DDIT3* significantly reduced the expression of Gln metabolic genes in WT cells (*Figure 3D*). Established Myc targets involved in glycolysis, lipid, or nucleotide metabolism were not significantly affected by 4EKI (*Figure 3—figure supplement 1G*, *Supplementary file 1B*).

We next compared the growth of 4EKI and parental cells in vivo. 4EKI clones showed marked reduction in engraftment efficiency (30–50%) and growth rate (over 90%) in nude mice (*Figure 3E–F*). 4EKI tumors were very small and showed near complete loss of p-4E, p-4E-BP1 (S65/T70), with drastic reduction in Myc, proliferation (Ki-67), ISR (p-GCN2, p-eIF2a) and AKT (S473) (*Figure 3G–H*), but marked stabilization of eIF4E and 4E-BP1 interaction (over 1000-fold) as measured by PLA (*Figure 3I–J*). p-GCN2 and p-eIF2a staining was punctate in xenograft tumors (*Figure 3—figure supplement 1H*). The expression of *EIF4E* was unchanged in 4EKI tumors, while that of *ATF4, DDIT3, MYC* was moderately reduced. The expression of Gln metabolic and translational targets was more severely reduced (by 60–90%) (*Figure 3K*). Among them, glutaminase (GLS) and leucine transporter SLC7A5 are key regulators of glutamine-dependent biosynthesis and mTOR activation (*Palm and Thompson, 2017*). Cell death (TUNEL), p-S6, or *EIF4E* was unchanged in 4EKI tumors with a minor reduction in several apoptotic targets such as *TNFRSF10B* (DR5) and *PMAIP1* (NOXA) (*Figure 3G*, *Figure 3—figure supplement 1I–J*). The expression of Gln metabolic and apoptotic targets (*SLC1A5, SLC7A5, BBC3, TRIB3* and *PMAIP1*) is also elevated in TCGA CRC cohorts (*Figure 3—figure supplement 1K*). These data support that p-4E promotes Myc- and ATF4-driven CRC growth through chronic exploitation of the most ancient arm of ISR (GCN2) to maintain constitutive AKT/4E-BP1 signaling.

## eIF4E S209 promotes Myc- and ISR-driven tumor initiation and progression

*Apc^{Min/+}* mice are a widely used model for Myc-driven cancer initiation and polyposis (*Qiu et al., 2010*; *Leibowitz et al., 2014*). Polyps developed in *Apc^{Min/+}* mice showed highly elevated p-4E, p-4E-BP1 (S65/T70), Myc, p-AKT (S473) and p-eIF2α, compared to highly proliferative (Ki-67+) adjacent 'normal' crypts (*Figure 4A*). p-eIF2α and p-AKT (S473) staining was punctate in the polyps and largely absent in the crypts (*Figure 4A*). Mice homozygous for the *Eif4e^{S209A}* allele (*4EKI/KI*) develop normally (*Furic et al., 2010*). Crossing *4EKI (S209A)* allele onto *Apc^{Min/+}* mice significantly reduced tumor burden in a dose-dependent manner (*Figure 4B–C*, *Figure 4—figure supplement 1A–B*). Intestinal analysis revealed uniform p-4E staining in the proliferating zone (Ki-67+) of the crypts in WT but not 4EKI *Apc^{Min/+}* mice (*Figure 4—figure supplement 1C*). However, 4EKI had no effect on intestinal proliferation (Ki-67), or the expression of *Eif4e, Myc,* or *Atf4*, while significantly reduced the expression of *Ddit3,* and several metabolic, translational, and apoptotic ISR targets (*Figure 4—figure supplement 1D–F*). In contrast, 4EKI polyps showed no p-4E staining, reduced p-4E-BP1, Myc and p-AKT (*Figure 4D–E*), with little effect on *Myc* mRNA, Ki-67, or p-S6 compared to WT polyps at similar sizes (*Figure 4F*, *Figure 4—figure supplement 1G–H*). However, a potential role of p-4E in non-epithelial compartments or p-4E-independent mechanisms cannot be ruled not. We also found that p-4E, not total eIF4E, is highly elevated in human adenomas (*Figure 4G*), consistent with highly elevated Myc protein in these precursor lesions and modestly elevated mRNA (*He et al.,*

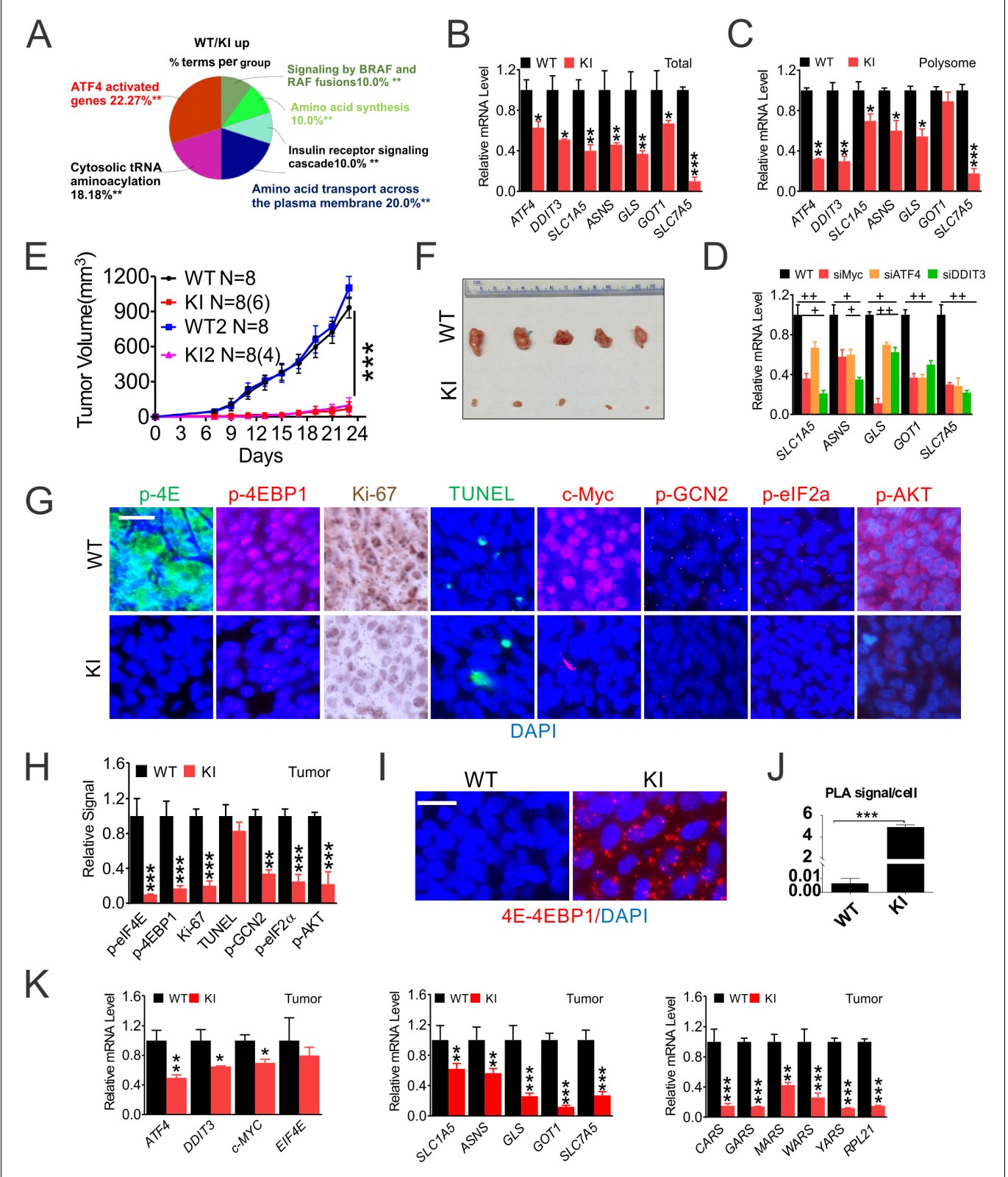

**Figure 3.** eIF4E S209 regulates CRC growth and the ISR in vivo. Paired HCT116 WT and 4EKI cells were characterized in vitro and in vivo. (**A**) Reactome Pathway analysis of genes elevated in WT cells (3-fold or more) visualized by ClueGO. (**B**) RT-PCR of indicated transcripts in total and (**C**) in polysomal RNAs. (**D**) The indicated transcripts were analyzed by RT-PCR in HCT116 cells transfected with the indicated siRNAs for 24 hr. N = 3. (**E–K**) Xenograft tumors. (**E**) The growth curve of WT and KI (KI1 and KI2) cells (four million) injected s.c. into different flanks of nude mice by day 23. (**F**) Representative

*Figure 3 continued on next page*

*Figure 3 continued*

images of WT and KI (KI1) xenografts on day 23. (**G**) Representative IF/IHC of the indicated makers in randomly chosen WT and KI tumors. Scale bar: 50 μm and (**H**) quantification of markers in (**G**) using 400x fields. (**H**) Representative PLA of eIF4E-4E-BP1 binding, scale bar: 25 μm, and (**J**) quantification (dots/cell). (**K**) The indicated transcripts were detected by RT-PCR. WT values were set at 100%. (**B-D, F, H, J-K**), values represent mean+s.d. (n = 3 or as indicated). *p<0.05, **p<0.01, ***p<0.001 (Student's t-test, two tailed). WT vs. KI. ⁺p<0.05, ⁺⁺p<0.01, (one-way ANOVA with TUKEY post-hoc test), or scrambled vs. specific siRNAs.

The online version of this article includes the following figure supplement(s) for figure 3:

**Figure supplement 1.** eIF4E S209 regulates the ISR in vitro and in vivo.

*1998*). These data establish that p-4E is a rate-limiting factor in Myc- and ISR-dependent AKT signaling and oncogenic proliferation.

## eIF4E S209 promotes Myc- and ISR-dependent glutamine addiction

The above and the effects of Mnki on CRC organoids (*Figure 2I–J*) suggest that p-4E-dependent increase in Myc and glutamine metabolism might serve as a druggable vulnerability. Both glucose and Gln were required for the growth of HCT116 cells in culture, while 4EKI reduced cell loss and apoptosis upon deprivation of Gln, not glucose (*Figure 5A–C*). In WT cells, cell death and caspase-3 cleavage from 24 to 48 hr followed a rapid loss of p-4E/4E-BP1 (S65/T70) and Myc, and increased activation of ISR (p-GCN2/p-eIF2a and ATF4/CHOP) by 4 hr. The levels of apoptotic targets (DR5 and PUMA) decreased transiently by 4 hr and strongly increased by 24 hr (*Figure 5D*), indicative of failed recovery with increased cell death. In contrast, 4EKI cells showed reduced basal levels of Myc, ISR and p-AKT, transient ISR induction. Reduced or abrogated induction of ATF4, CHOP and apoptotic targets by 24 hr was consistent with a more successful recovery (*Figure 5D*). The kinetics or levels of p-4E-BP1 (T37/46), p-ERK, p-mTOR (S2448), p-S6, p-PERK, cyclin D1 were not significantly affected by 4EKI, while feedback AKT activation at 24 hr was observed in both cell lines and higher in WT cells (*Figure 5D*).

RT-PCR analysis using polysomal and total cellular RNAs further revealed coordinated transcription and translation during the ISR. *MYC* loss and *ATF4* induction was rapid and primarily in polysomal fractions, while increases in *CHOP* and downstream effectors (metabolic and apoptotic) were in both fractions (*Figure 5E–F*). Knockdown of *MYC*, *ATF4*, and to lesser extent, *DDIT3*, *SLC1A5*, or *ASNS*, reduced cell death along with the basal and ISR induction in WT cells upon Gln deprivation (*Figure 5F–G*, *Figure 5—figure supplement 1A*). Compared to the increase in ISR effectors, the reduction of *MYC* was minimal in the total RNAs by Gln deprivation, 4EKI, or knockdown of *ATF4* or *DDIT3*, confirming a key role of its posttranslational regulation (*Figure 5F*). Conversely, overexpression of *MYC* or *ATF4* increased the ISR and apoptosis in 4EKI cells (*Figure 5H*, *Figure 5—figure supplement 1B*). The sensitivity to Gln deprivation was dose-dependent in parental cells (*Figure 5—figure supplement 1C–D*) and blunted in the independent 4EKI2 clone (*Figure 5—figure supplement 1E–G*). In addition, 4EKI cells were less sensitive than parental cells to GLS inhibitor CB-839 with reduced apoptosis and ISR induction (*Figure 5I*, *Figure 5—figure supplement 1H–I*). Together, these findings establish p-4E/Myc in promoting glutamine addiction in CRC cells via ISR hyperactivation and failed recovery upon withdraw.

## Mutant *KRAS* cooperates with p-4E/Myc to promote glutamine addiction and immune suppression

*KRAS* mutations are frequently found after *APC* inactivation in large colonic adenomas (*Vogelstein et al., 2013*). Established CRC lines including HCT 116 often harbor mutant *KRAS* or *BRAF* (*Supplementary file 2*), which leads to constitutive ERK/MAPK signaling and elevated p-4E (*Ueda et al., 2004*; *Ueda et al., 2010*). We then determined if mutant *KRAS* contributes to glutamine addiction via p-4E/Myc and the ISR. Isogenic CRC cells (HCT116, Lim1215 and SW48) harboring mutant *KRAS* (G13D or G12V) showed higher basal levels of p-ERK, p-4E/4E-BP1, p-eIF2α and Myc, and are more sensitive to Gln deprivation, compared to their *KRAS* wildtype counterparts (*Figure 6A–B*). The increased sensitivity was associated with elevated apoptosis and ISR, and reduced Myc, p-4E but not p-ERK (*Figure 6C–E*, *Figure 6—figure supplement 1A–C*).

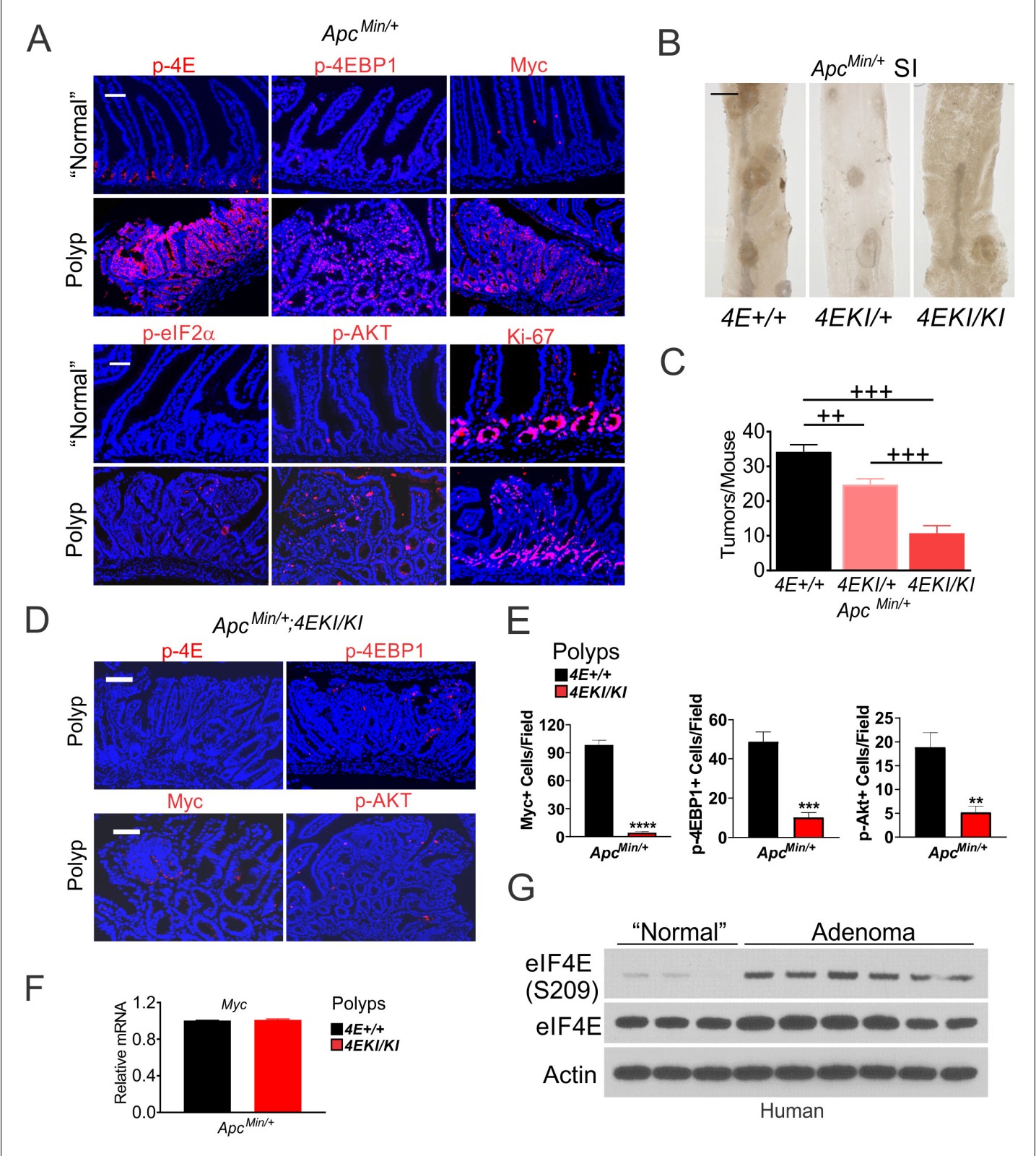

**Figure 4.** eIF4E S209 promotes Myc-driven oncogenesis in mice and human. (**A**) Representative IF of indicated markers in polyps and adjacent 'normal' crypts in $Apc^{Min/+}$ mice. Scale bar: 100 µm. (**B**) Representative images of polyps and (**C**) quantification in the small intestine of 3-month-old $Apc^{Min/+}$ (4E +/+) (n = 5), $Apc^{Min/+}$;4EKI/+ (n = 8), and $Apc^{Min/+}$;4EKI/KI (n = 4) mice. $^{++}p<0.01$, $^{+++}p<0.001$ (One-Way ANOVA with TUKEY Post-hoc test). (**D**)

*Figure 4 continued on next page*

*Figure 4 continued*

Representative IF of indicated markers and (**E**) quantitation in the polyps in indicated mice. Scale bar: 100 μM. (**F**) Myc mRNA levels in the polyps from indicated mice. (**G**) p-4E (S209) and total eIF4E in human adenomas and normal colon analyzed by western blotting. Actin is a loading control.

The online version of this article includes the following figure supplement(s) for figure 4:

**Figure supplement 1.** eIF4E S209 promotes polyposis *in Apc^{Min/+}* mice.

We further analyzed transcriptomic landscapes to better understand the role of p-4E in the response of mutant *KRAS* to Gln deprivation. 4EKI significantly reduced the overall transcriptional response induced at either direction by nearly 50%, and drastically altered target selection (2-fold or more, *Figure 6F–G*, *Figure 6—figure supplement 1D*). Only a minor fraction of induced genes (117) (22% and 35%, respectively) was shared and mapped to stress and ATF4-related pathways (Top five non-overlapping) (*Figure 6H*). WT-specific induced genes (409,~78%) were mapped to transcription, cell cycle and cell death (*Figure 6I*), while 4EKI-specfic ones (217,~65%) were mapped to immune pathways (*Figure 6J*). These data support that p-4E critically controls the ISR level and outcomes such as cell death, transcriptional targets, and immune suppression in mutant *KRAS* CRC cells upon glutamine deprivation.

## p-4E controls cell death and transcriptomic heterogeneity upon metabolic stress

We used a panel of CRC lines with diverse driver mutations (DLD1, RKO, SW480, and HT29) (*Supplementary file 2*), to further determine if elevated p-4E is associated with Gln addiction and transcriptional heterogeneity. The increased sensitivity was associated with higher basal p-4E/p-4E-BP1 and Myc, and enhanced ISR induction and reduced p-4EBP1 and Myc reduction upon deprivation, with little or no effect on the levels of p-ERK, p-4E, or p-S6 (*Figure 7A–C*). Cell death was correlated with the induction of p-GCN2/p-eIF2a, ATF4/CHOP and cleaved caspase-3 in RKO and H29 cells (*Figure 7—figure supplement 1*). We then analyzed induced genes in three CRC lines (HCT 116, RKO, and HT29 (MSI or MSS with mutant *KRAS* or *BRAF*)) sensitive to Gln deprivation. Among nearly 2000 induced genes (2-fold or higher), 100–200 (5–10%) were shared by any two cell lines. Only 53 genes (~4%) were shared by all three cell lines and mapped to ATF4, stress and angiogenesis (VEGF) (*Figure 7D–E*). Together, our data support that metabolic stress selectively triggers ISR-dependent cell death in p-4E/Myc high cells, which is associated with marked transcriptional heterogeneity (*Figure 7F*).

## Discussion

Translational reprogramming has emerged as a key regulator of cancer development, and the underlying mechanisms remain not fully resolved (*Truitt and Ruggero, 2016*; *Robichaud et al., 2015*; *Robichaud et al., 2018*). Our study demonstrates that p-4E-mediated Myc and ATF4 translation cooperates with mutant *KRAS* to promote CRC cell growth through ISR-dependent glutamine metabolism and AKT/p-4E-BP1 signaling (*Figure 7F*). High levels of p-4E and p-4E-BP1 (S65/70) are likely required to sustain Myc and ATF4 translation in cancers such as CRC with rare *MYC* gene rearrangements or amplifications (*Vogelstein et al., 2013*; *Fan et al., 2009*; *He et al., 1998*) and prominent increase in ribosomal biogenesis (*Truitt and Ruggero, 2016*; *Zhang et al., 1997*). These findings reveal fundamental differences between normal and oncogenic translation and proliferation. p-4E serves as an early and rate-limiting step in Myc- and stress-driven oncogenesis through GCN2-selective amino acid sensing and AKT activation. p-4E-dependent increase in Myc or AKT signaling is largely dispensable for rapid intestinal proliferation under homeostatic conditions.

Our data support a powerful and translational mechanism in shaping driver and TME interactions and evolution under metabolic stress (*Figure 7F*; *Robichaud and Sonenberg, 2017*). This finding is consistent with the lack of metastasis-specific mutations or driver heterogeneity in treatment-naïve CRCs (*Jones et al., 2008*; *Reiter et al., 2018*). The ISR is intrinsically redundant, heterogeneous, and crosstalk with mTOR signaling (*Hetz et al., 2013*; *Tabas and Ron, 2011*; *Appenzeller-Herzog and Hall, 2012*), which is increasingly recognized as an engine of 'non-oncogenic addiction' (*Tameire et al., 2015*; *Cubillos-Ruiz et al., 2017*; *Luo et al., 2009*). The levels of p-4E/4E-BP,

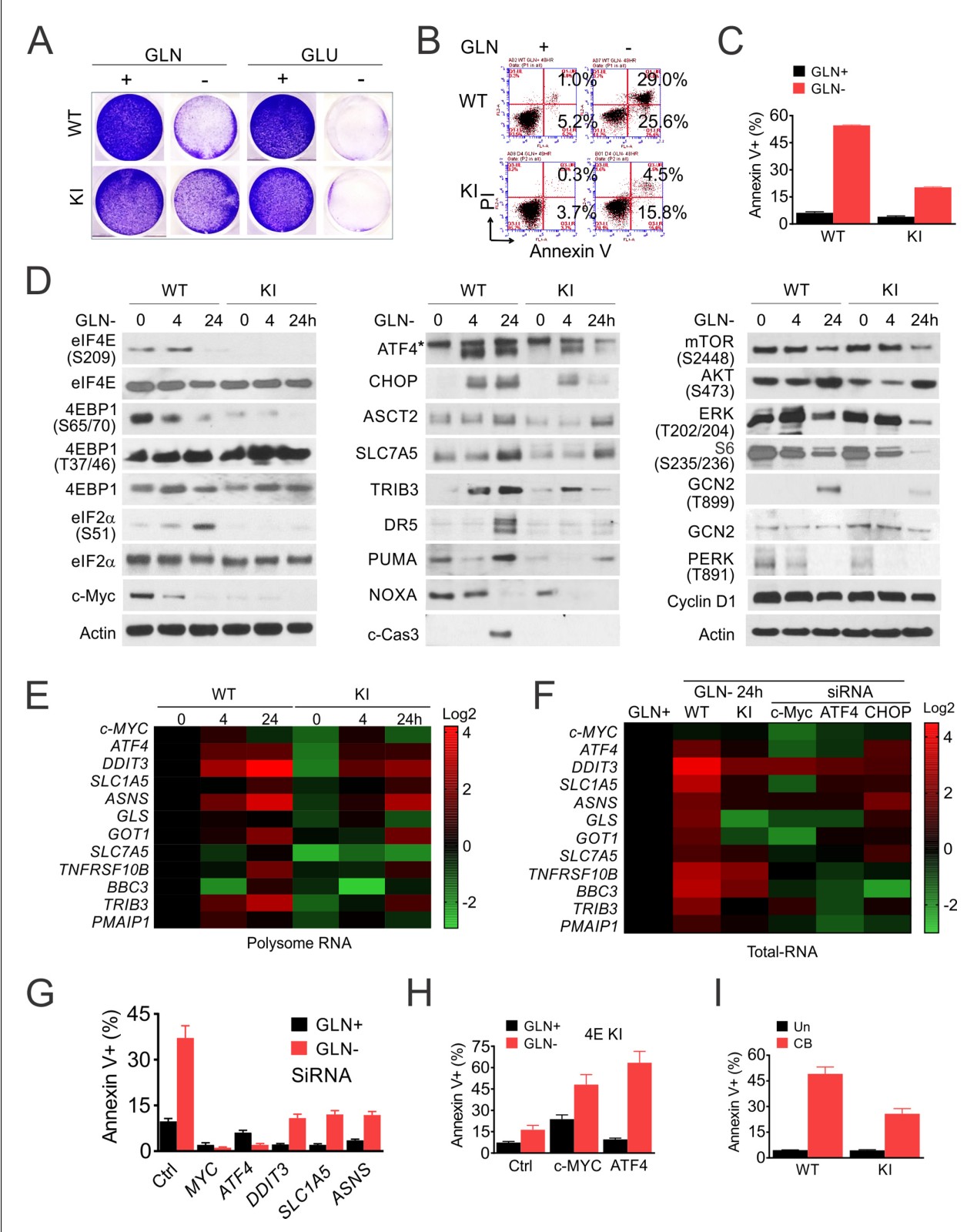

**Figure 5.** p-4E promotes ISR-dependent glutamine addiction. Isogenic HCT 116 WT and 4E KI cells were subjected to glutamine deprivation (2 to 0 mM), glucose deprivation (4 to 0 mM) or GLS inhibitor (CB-839, 40 µM) treatment, and analyzed at indicated times (0–48 hr). (A) Attached cells were stained with crystal violet at 48 hr. (B) Cell death were analyzed at 48 hr by flow cytometry and (C) quantification of AnnexinV+ cells. Representative flow cytometry plots and results are shown. (D) The indicated proteins were analyzed by western blotting at indicated times after Gln deprivation. (E)

*Figure 5 continued on next page*

Figure 5 continued

Heatmap of indicated transcripts in polysomes at 24 hr were analyzed by RT-PCR, normalized to WT 0 hr (*Siegel et al., 2019*). (F–G) WT cells were transfected the indicated siRNAs for 24 hr, followed by 24 hr recovery, and subjected to glutamine deprivation. (F) Heatmaps of indicated transcripts in total RNA at 24 hr were analyzed by RT-PCR, normalized to WT 0 hr (*Siegel et al., 2019*). (G) Cell death at 48 hr was quantitated by Annexin V+ cells. (H) 4EKI cells were transfected with control, Myc or ATF4 expression construct and subjected to glutamine deprivation. Cell death at 48 hr was quantitated by Annexin V+ cells. (I) Cell death 48 hr after GLSi was quantitated by Annexin V+ cells.

The online version of this article includes the following figure supplement(s) for figure 5:

**Figure supplement 1.** p-4E drives Myc and ISR-dependent glutamine addiction.

p-eIF2α, and p-AKT are heterogenous in the TME (*Figure 4*; *Le Bacquer et al., 2007*). Our data showed that mutant *KRAS* cooperates with p-4E and Myc to promote ISR-dependent Gln addiction, which is characterized by increased cell death, transcriptional heterogeneity, and surprisingly immune suppression upon deprivation. Hyperactivation of Wnt/Myc and mutant *RAS/RAF* are strongly associated with cancer aggressiveness including immunosuppressive TME, absence of TILs, and resistance to immune checkpoint inhibitors (*Ruan et al., 2020*). Our data provide direct evidence that p-4E rewired translation and stress response is involved in this process through increased entropy or chaos (*Tarabichi et al., 2013*; *Hanselmann and Welter, 2016*) and target selection during chronic metabolic adaptation and competition within tumor cells and with the TME (*Palm and Thompson, 2017*; *Andrejeva and Rathmell, 2017*). Our findings are consistent with the notion that p-4E is linked to increased aggressiveness through the translation of immune checkpoint PD-L1 (*24*) and metastasis targets such as SNAIL and MMP3 (*30*) in Myc, mutant KRAS or PI3K-driven cancer models. ISR-dependent and sustained metabolic (Gln) and translation (tRNA synthetases) (*Han et al., 2013*) pressure can lead to cell death, while transient pressure likely permits better cell survival and Myc recovery (*Dejure et al., 2017*). Therefore, nutrient- and p-4E-sensitive translation is predicted to dynamically regulate the extent and heterogeneity of tumor intrinsic ISR, TME interaction and outcomes, which is worth further exploring (*Figure 7F*).

Our study helps shed some light on the biochemical nature of oncogenic translation and supports a 'coded' model, in which elevated p-eIF4E/4E-BP and p-eIF2α chronically 'stress' translation toward targets such as Myc, ATF4 and MMP7. This 'coded' model is supported by dynamic changes in their phosphorylation and binding under stress, target selection coupled with transcription, and dynamic 4F and target interactions (*Costello et al., 2017*). This model is also in line with the notion that 'reprogrammed' translation serves specialized roles including immunity (*Pelletier et al., 2015*; *Truitt and Ruggero, 2016*; *Xu et al., 2019*; *Harding et al., 2000*), and regulatory roles of extensive phosphorylation sites in 4E-BP1 (T37/T46, S65/T70, S83, S101, and S112) and family members (*Martineau et al., 2013*). Punctate staining of p-GCN2 and p-eIF2a and 4E-4EBP1 interaction in xenografts also supports transient and localized GCN2-eIF2a regulation (*Castilho et al., 2014*) with dynamic translation control during tumor evolution (*Figure 7F*). High p-4E levels render 4E-BP1 and Cap binding insensitive to serum or growth factors (*Figure 2D*). It will be interesting to determine whether and how these 'trans' codes might work with various 'cis' elements in mRNAs to control the extent and levels of inducible translation in the context of oncogenic drivers. Careful dissection of site-specific and conditional 4E/4E-BP mutants using appropriate in vivo models will help better understand how stress maladaptation leads to cancer and other chronic diseases (*Tameire et al., 2015*; *Cubillos-Ruiz et al., 2017*; *Luo et al., 2009*) at the systems level.

Myc, mutant RAS/RAF, or PIK3CA fuels abnormal growth by altering nutritional needs, including increased glutamine utilization. However, direct targeting a single driver, a single metabolic step, p-4E, or p-4E-BP1 has limited success in the clinic so far, due to complex resistance mechanisms encompassing compensatory pathway activation, metabolic bypasses, apoptosis resistance and immune suppression (*Dang et al., 2017*; *Pelletier et al., 2015*; *Truitt and Ruggero, 2016*; *Palm and Thompson, 2017*; *Ruan et al., 2020*; *Lito et al., 2013*; *Zhang and Yu, 2013*). Our model supports that Myc-driven metabolic addiction likely goes beyond Gln to other nutrients and reducing agents such as arginine, cystine, and glutathione necessary to support increased biosynthesis (*Figure 7F*). Elevated ISR in such tumors might therefore offer a therapeutic vulnerability. In fact, CRCs harboring mutant *RAS/RAF* or *SPOP* can be more effectively killed by drug combinations through catastrophic ISR associated with features of immunogenic cell death (*He et al., 2013*;

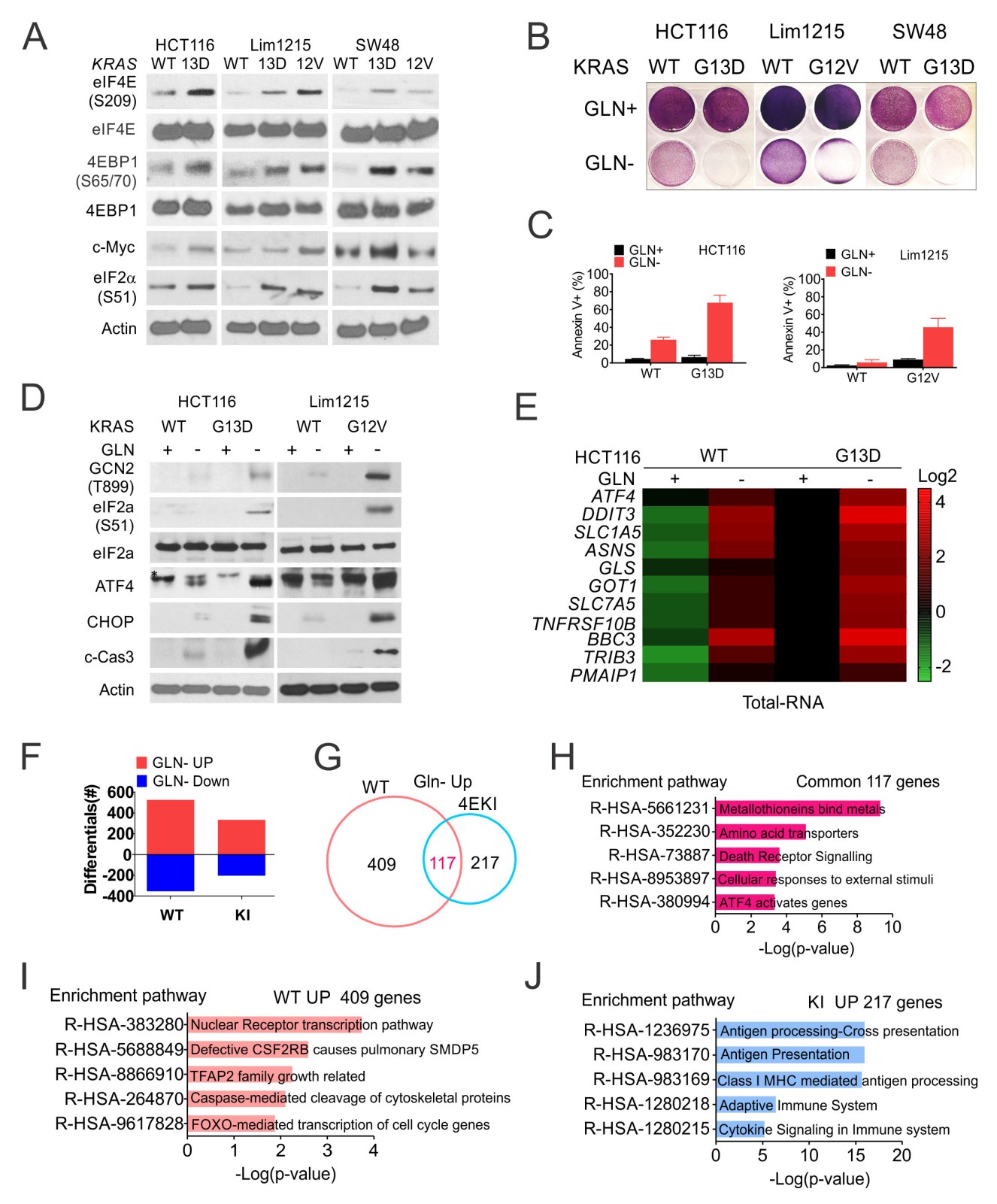

**Figure 6.** Mutant *KRAS* and Myc promotes p-4E-dependent glutamine addiction. Indicated WT and mutant *KRAS* isogenic CRC lines were subject to glutamine deprivation and analyzed. (**A**) The basal levels of indicated proteins were analyzed by western blotting. (**B**) Attached cells at 48 hr were stained with crystal violet. (**C**) Cell death at 48 hr was quantitated by flow cytometry. Representative results were shown. (**D**) The indicated proteins at 24 hr in two pairs of cell lines were analyzed by western blotting. *, non-specific band. (**E**) Heat map of indicated transcripts at 24 hr were detected by RT-

*Figure 6 continued on next page*

Figure 6 continued

PCR and normalized to the control (WT, Gln+, 1). (F) Numbers of differential genes upon Gln deprivation in the isogenic HCT 116 pair, 2-fold or more, *p<0.05. (G) Venn diagram of induced genes from (F). (H–J) Reactome pathway analysis of (H) shared, WT (I), and 4EKI-specific upregulated genes(J). Top five enriched pathways are shown.

The online version of this article includes the following figure supplement(s) for figure 6:

**Figure supplement 1.** Mutant *KRAS* promotes glutamine addiction.

He et al., 2016a; He et al., 2016b; Li et al., 2017; Tan et al., 2019), which is believed to help achieve more durable cancer control in patients (Ruan et al., 2020). The use of in-treatment and patient-derived models (Silva-Almeida et al., 2020) will likely be important to capture dynamic transcriptional responses and biomarkers for tailoring therapeutic inventions.

In summary, we demonstrate that p-4E-rewired translation is essential for Myc- and stress-driven oncogenesis. These findings provide a more unified mechanism to explain how cancer hallmarks are continuously shaped by drivers and the TME. Elevated p-4E and ISR is a potential therapeutic vulnerability in CRCs that are dependent on high levels of Myc and mutant *KRAS*.

## Materials and methods

### Cell culture and treatment

The human CRC cell lines, including HCT116, DLD1, RKO, SW480, and HT29 were obtained from the American Type Culture Collection (Manassas, VA, USA). Isogenic *KRAS* pairs HCT116 (WT, G13D), Lim 1215 (WT, G13D, G12V), and SW48 (WT, G13D, G12V) cell lines were obtained from Bert Vogelstein at Johns Hopkins University as described (Yun et al., 2009). Information on major drivers or isogenic cell lines are found in (Supplementary file 2).

Authentication of purchased cell lines was provided by ATCC. Genetically modified cell line pairs were additionally verified by published biomarkers compared with parental (unmodified ATCC parental lines) as well as DNA sequencing and western analysis. Cell lines were regularly monitored and confirmed to be absence of Mycoplasma. All cell lines were used for less than 2 months (10 or fewer passages) in culture upon thawing from LN tank. All cell lines were cultured in McCoy's 5A modified medium (Invitrogen, Carlsbad, CA, USA, Cat# 16600–082) supplemented with 10% defined fetal bovine serum (Hyclone, Logan, UT, USA, Cat # SH3007103), 100 units/ml penicillin, and 100 µg/ml streptomycin (Invitrogen) unless noted otherwise. Cells were maintained in a 37°C incubator at 5% CO2. mRNA or protein was analyzed at 24 hr while cell growth or death at 48 hr, unless noted otherwise.

### $EIF4E^{S209/+}$ knockin human CRC cells

The $EIF4E^{S209A}$ (4EKI) targeting vector was constructed using the pUSER-rAAV (recombinant adeno associated virus) System (Zhang et al., 2008). Briefly, 2 0.8 kb homologous arms flanking the seventh intron of eIF4E were inserted between 2 USER sites in the AAV shuttle vector pTK-Neo-USER. The coding sequence for eIF4E single mutant (S209A) was introduced into the right arm using the QuickChange XL Site-Directed Mutagenesis Kit (Agilent Technologies, Cat# 200516). For gene targeting, HCT116 cells were infected with the targeting rAAV and selected by G418 (0.5 mg/ml; Mediatech) for 3 weeks. G418-resistant clones were pooled and screened by PCR for targeting events. To target the second allele, Neo flanked by 2 LoxP sites was excised from a heterozygous clone (209S/209A) by infection with an adenovirus expressing Cre recombinase (Ad-Cre) followed by isolation of single clones. The same targeting construct was used in the second round of gene targeting. The 4EKI targeting in independent HCT 116 clones was verified by sequencing of genomic DNA and Western blotting. We were unable to obtain or recover any homozygous 4E KI (209A/209A) HCT 116 cells after three rounds (3–4 T75s/each round, ~1 billion cells in total) with only retargeting events of the 'Cred' 4E KI allele. We were unable to obtain or expand any 4E KI (209S/A) heterozygous RKO or HT29 clones (p-4E high) from multiple PCR positive pools identified initially in 96-well plates. Details on screening and validation primers are found in (Supplementary file 3A).

Colony formation assay. 600 cells were plated in 6-well tissue culture plates. Cells were maintained at 37°C/5% CO2 and allowed to grow for 10 days with media changed every other day. At the

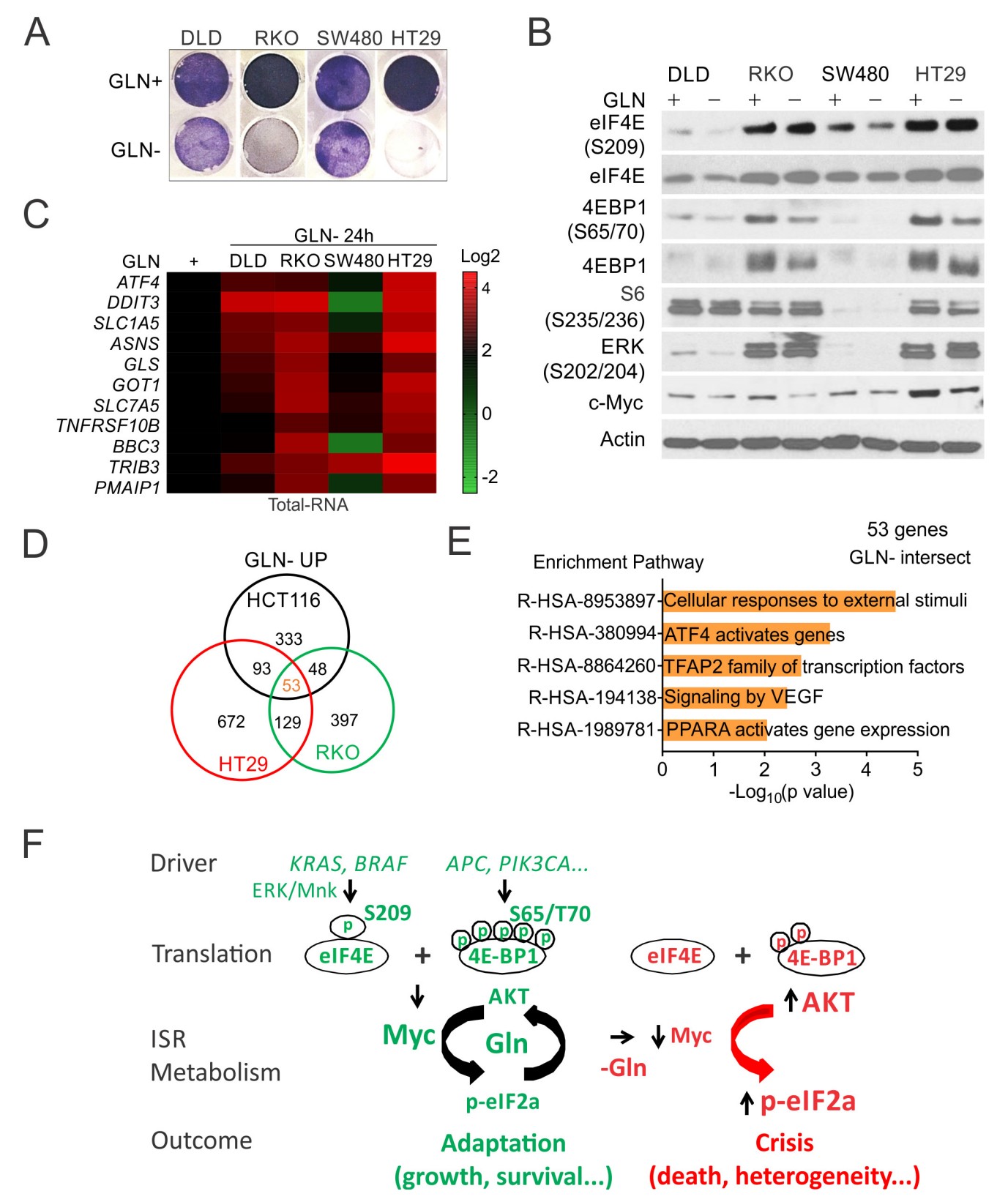

**Figure 7.** p-4E controls metabolic stress-induced cell death and transcriptomic heterogeneity. Indicated CRC lines were subject to glutamine deprivation and analyzed. (**A**) Attached cells at 48 hr were visualized by crystal violet. (**B**) The indicated proteins were analyzed by western blotting. (**C**)

*Figure 7 continued on next page*

*Figure 7 continued*

Transcripts at 24 hr were analyzed by RT-PCR and normalized to respective cell line controls (Gln+, 1). **(D)** Venn diagram of Gln deprivation induced genes in three sensitive cell lines. Two fold or more, *p<0.05. **(E)** Reactome Pathway analysis of shared genes (*Silva-Almeida et al., 2020*) in **(D)**. **(F)** A model of p-4E in CRC development. CRC drivers converge on increased p-4E (S209) and p-4E-BP1(S65/T70) to promote Myc- and ISR (p-eIF2a/ATF4)-dependent adaptation and AKT activation via increased glutamine (Gln) metabolism. Acute metabolic stress (i.e. Gln deprivation) disrupts adaptation and triggers crisis in such cells due to rapid loss of p-4E/4E-BP1 and Myc and hyperactivation of ISR and p-AKT, leading to increased cell death and transcriptional heterogeneity.

The online version of this article includes the following figure supplement(s) for figure 7:

**Figure supplement 1.** Glutamine deprivation leads to ISR hyperactivation in mutant *KRAS/BRAF* CRC cells.

conclusion of 10 days, cells were washed once with HBSS before staining with Crystal violet dye for 10 min. Cells were then washed three times with HBSS and photographed. All conditions were performed in triplicate.

Soft agar assay. Agar plates were prepared in 6-well petri dishes by first applying a 1.5 ml base layer of 1.6% agar (Invitrogen, Cat# 16500500) in Dulbecco's modified Eagle's medium containing 10% fetal bovine serum. Over this basal layer, an additional 1.5 ml layer of 0.8% agar in the same medium/fetal bovine serum mixture, and 1,000 cells were added. The cells were incubated at 37°C in a humidified atmosphere of 5% $CO_2$ in air and gently layer 2 ml of 1x media over top agarose containing cells next day. 2 weeks later, colonies were measured unfixed and unstained by using a microscope with a calibrated grid. Clones with greater than 50 cells were scored as positive.

Spheroid (3D) culture. Cells were plated into 96-well round–bottom ultralow attachment plates (Greiner Bio-one #650970) at densities of 100–3000 cells/well in 200 µl/well McCoy's 5A modified medium (Invitrogen, Carlsbad, CA, USA, Cat# 16600–082) supplemented with 10% defined fetal bovine serum (Hyclone, Logan, UT, USA, Cat# SH3007103), 100 Penicillin-Streptomycin (10,000 U/ml) (Invitrogen #15140–122). Plates were briefly centrifuged at 250 g for 5 min and then incubated at 37°C and 5% CO2. Spheroids formed within approximately 24 hr. Cells were re-fed every 72 hr by carefully removing 100 µl of medium from each well and replenishing with 100 µl of fresh growth medium. The growth of spheroids was imaged at indicated times (3–7 days) using an Olympus BX51 system microscope equipped with SPOT camera and SPOT Advanced 5.1 software.

## Analyses of cell viability, proliferation and cell death

Cell counts. 50, 000 cells were plated in triplicate in each of a 12-well tissue culture plate. Following plating, cells were harvested at day 1, day 3, day 5, and day 7. Cells were trypsinized and counted using a hemacytometer.

Cell viability was measured by CellTiter 96 AQueous One Solution Cell Proliferation Assay (MTS) (Promega G3580), and ATP-based CellTiter-Glo Luminescent Cell Viability Assay Kit (Promega, G7570) was performed in 96-wells according to the manufacturer's instructions. The absorbance at 490 nm and Luminescence was measured with a Wallac Victor 1420 Multilabel Counter (PerkinElmer, SKU#: 8381-30-1005). Following various treatments, attached cells or clones were stained and with crystal violet (Sigma, St. Louis, MO, Cat# C0775) (3.7% Paraformaldehyde, 0.05% crystal violet in distilled water and filtered at 0.45 um before use) (*Yu et al., 2003*).

Bromodeoxyuridine (BrdU) incorporation was analyzed in cells following 15 min BrdU (10 µM) pulse. Cells were washed three times with Hank's following fixation with methanol: acetone (1:1) at −20°C for 15 min, and staining with a monoclonal Alexa Fluor 594 anti-BrdU conjugated antibody (Invitrogen, Cat# B35132) and DAPI, and scored under a fluorescence microscopy as described (*Wang et al., 2009*).

Cell death and apoptosis were analyzed by nuclear staining with cells harvested from 12-well plates and Hoechst 33258 (Invitrogen, Cat# 40045), and Annexin V/propidium iodide (PI) followed by flow cytometry as described (*Yu et al., 2006*). Experiments were repeated on two or more occasions (different days) with similar results. Flow cytometry plots and quantitation were based on the analysis of 20,000 cells for each condition. Results from one representative experiment are shown with fraction (%) of indicated population.

Glutamine deprivation. Cells were plated overnight in complete DMEM (10% FBS, 2 mM glutamine) briefly washed with phosphate-buffered saline (PBS) (Gibco #14175–095) and then transferred

into glutamine-free medium (glutamine- and pyruvate-free DMEM (Corning Cellgro Cat# 15–017-CV)) supplemented with 10% dialyzed FBS (Gibco A3382001) and 1 mM sodium pyruvate (Gibco #11360070). The corresponding glutamine-replete medium was prepared by addition of 2 mM glutamine (Corning Cellgro #25–005 CI) to glutamine-free medium. In some experiments, cells were first transfected with siRNA or expression constructs for 24 hr and replated in complete medium 24 hr to reach 30 to 40% density before the treatment.

Drug treatment. Cells were plated in 12-well plates 24 hr before and reached 30 to 40% density at treatment. The chemicals used include 40 µM CB839 (MCE, HY-12248), All chemicals used were dissolved with DMSO (Sigma-Aldrich Cat # 276855) in the stock and diluted by culture medium with final DMSO concentration at or below 0.5%.

## Real-time Reverse Transcriptase (RT) PCR
Total RNA was isolated from cells using the Mini RNA Isolation II Kit (Zymo Research, Orange, CA) according to the manufacturer's protocol. One µg of total RNA was used to generate cDNA using Superscript III reverse transcriptase (Invitrogen, Carlsbad, CA, USA). Real-time PCR was carried out as described with triplicates normalized to WT or untreated (*He et al., 2013*). Details on primers are found in the supplementary materials (*Supplementary file 3B-C*). Representative results are shown, and similar results were obtained in at least three independent experiments. cDNA was synthesized from RNA prepare from cells, or RNA pooled from 2 to 3 randomly chosen tumors in each group.

## Western blotting
Western blotting was performed as previously described (*Yu et al., 2001*). Details on antibodies were found in the supplemental materials (*Supplementary file 3D*).

## Polysome fractionation
HCT116 parental and 4EKI cells were maintained in high-glucose (4.5 g/L) DMEM media with 10% FBS, 100 µg/ml penicillin and 100 µg/ml streptomycin. Early passage cells (within four after thawing) were plated at 30–40% and allow to grow to approximately 80% confluence within two days, cells were treated with 100 µg/ml cyclohexamide (Sigma CAS:66-81-9) in fresh cell culture medium at 37°C for 10 min. Cells from two 10 cm-plate were scraped and incubated in 600 µl lysis buffer (20 mM Tris-HCl pH8, 140 mM NaCl, 1.5 mM MgCl$_2$, 0.25% NP-40, 1% Triton-X 100, 10 mM DTT, 200 µg/ml cyclohexamide and 200 U/ml Rnasin) for 30 min on ice. Lysates were centrifugated at 12000 rpm for 10 min at 4°C and supernatants were loaded onto a 10–50% sucrose gradient with 1 mM DTT, 100 µg/ml cyclohexamide and 40 U/ml Rnasin. Samples were centrifugated at 40,000 rpm for 2.5 hr at 4°C (Beckman coulter, SW41 Ti rotor) and then separated on a BIOCOMP gradient fractionation system (Canada, BIOCOMP, Model 251) to evaluate polysome profiles and collect polysome fractions. RNA was isolated from polysome fractions using TRIpure LS Reagent (aidlab Cat# RN0202). Three sets of experiments were performed independently.

## m7GTP Pull-down assay
HCT116 parental and 4EKI cells were plated at 30–40% in T75s and allow to grow to 80% confluence within two days. Then cells were starved for 24 hr with Mycoy's 5A alone without FBS. After that replaced the cell media with fresh Mycoy's 5A without FBS for the control group or with 10% FBS for the stimulation group for another 4 hr. Immobilized gamma-Aminophenyl-7-methyl GTP (C10-spacer7-Methyl-GTP agarose beads from Amersham (Jena Bioscience, Germany, Cat# AC-155L)) was used for the pull-down assay using the method described previously with minor modifications (*Xu et al., 2010a*). In brief, 20 µl of m7 GTP agarose beads were washed with 500 µl of PBS three times, added to 300 µg of total protein from the cell lysates in 1 ml IP (RIPA buffer) lysis buffer and rotated overnight at 4°C. Cap-bound eIF4E and precipitates were washed 3 times with 500 µl IP lysis buffer and eluted using 100 µl protein sample buffer (1 M Tris pH6.8, 4% SDS, 20% Glycerol, 50% NaOH, and 10% β-Mercaptoethanol). The eluted proteins (20 µl) were analyzed by immunoblotting with 10% amount of input.

## Transfection

Transfection was performed using Lipofectamine 2000 (Invitorgen, Cat#11668019) according to the manufacturer's instructions and described (*He et al., 2016b*). The human pRK-ATF4 and pcDNA3.3-c-MYC constructs were obtained from ADDGENE. Cells were transfected with 0.4 µg of Plasmids/well in 12-well plates for 4 hr. *MYC, ATF4, DDIT3, SLC1A5* or *ASNS* small-interfering RNA (siRNA) duplexes were synthesized by Dharmacon (Lafayette, CO, USA). Cells were transfected with 400 pmols of siRNA duplexes/well in 12-well plates for 4 hr. Transfected cells after 4 hr were incubated in medium containing 5% FBS for 20 hr, replated in 10% FBS normal growth medium for 24 hr before treatment. Details for siRNA sequence and expression plasmids are found in (*Supplementary file 3E-F*) respectively.

Reporter assays. Cap-dependent luciferase reporter construct pcDNA-LUC (*Yang et al., 2004*) and bicistronic PIYC (*Nie and Htun, 2006*) were previously described. Cells in 12-well plate were transfected with 0.4 µg of the reporter and 2 µl of lipofectamine 2000 (Invitrogen, Cat#11668019). Cell lysates were collected 24 hr after transfections. Luciferase activities were measured and normalized to total protein level. All reporter experiments were performed in triplicate and repeated three times with similar results. HA- or Myc-tag reporters encoded by PIYC were analyzed by western blotting as described (*Xu et al., 2010b*).

## Animal studies

All animal experiments were approved by the University of Pittsburgh Institutional Animal Care and Use Committee. All methods were performed in accordance with the relevant guidelines and regulations. The protocols for the use of recombinant DNA and animals included IBC201700136, IACUC# 19085635 and 18063020.

## Xenograft studies

Female 5–6 week-old Nu/Nu mice (Charles River, Wilmington, MA) were housed in a sterile environment with micro isolator cages and allowed access to water and chow ad libitum. Mice were injected subcutaneously with $4 \times 10^6$ WT and 4EKI (KI1 or KI2) HCT116 cells on the opposite flanks of the same animal. Tumor growth was monitored by calipers, and tumor volumes were calculated according to the formula $0.5 \times \text{length} \times \text{width}^2$. Mice were euthanized when tumors reached 1.0 cm$^3$ in size in WT group, approximately 9 days after palpable WT tumors. Tumors were harvested at the end of experiments for analysis as described (*Yu et al., 2006*). Tumors were dissected and fixed in 10% formalin and embedded in paraffin or saved frozen for RNA extraction.

Immunohistochemistry (IHC) and immunofluorescence (IF). Rehydrated sections were treated with 3% hydrogen peroxide (IHC only), followed by antigen retrieval for 10 min in boiling 0.1 M citrate buffer (pH 6.0) with 1 mM EDTA. Apoptosis was analyzed by TUNEL staining with the ApopTag Peroxidase In Situ Apoptosis Detection Kit (Chemicon International, Temecula, CA) according to the manufacturer's instructions. Immunostaining was performed on 5 mm paraffin-embedded tumor sections using an Alexa Fluor 488- or Alexa Fluor 594-conjugated secondary antibody (Invitrogen) for detection. The details on the primary antibodies were found in (*Supplementary file 3D*).

Proximity ligation assay (PLA). Interactions of eIF4E and 4E-BP1 were detected by in situ PLA in paraffin-embedded sections. PLA was performed using the Duolink In Situ Redstarter Kit (Sigma Sigma-Aldrich Cat#DUO92101-1KT) according to the manufacturers' instructions and with minor modifications (*Chen et al., 2018*). Incubation with primary antibodies (*Supplementary file 3D*) was performed at 4°C overnight. The stained sections were mounted with VECTASHIELD Mounting Medium (Vector Laboratories) with DAPI for nuclear counter staining. The results were visualized by fluorescence microscopy (OlympusBX51 system), and the number of PLA signals per field was counted and plotted (more than three fields).

## *Apc*$^{Min/+}$ and *Eif4e*$^{S209A}$ mice

C57BL/6J *Eif4e*$^{S209A}$ homozygous (4EKI/KI) mice (*Furic et al., 2010*) were crossed with *Apc*$^{Min/+}$ mice (Jackson Laboratory) to generate *Apc*$^{Min/+}$ mice with different 4ES209 (209 S/S, 209S/A, or 209A/A) genotypes. Genotypes were verified by genomic PCR. For polyposis promotion, 4-wk-old mice were fed AIN-93G diets (Dyets) for two wks (~6 wk of age) for intestinal gene expression or two mo (12–13 wk of age) for tumor phenotype analysis. Following sacrifice, the small intestine and

colon were harvested for gene expression analysis and photographing and numeration of the adenoma burden as described (*Leibowitz et al., 2014*; *Qiu et al., 2009*). All measures were quantified from three randomly chosen mice. The RNA was pooled from three intestinal preps to produce cDNA for real-time PCR. In some cases, multiple adenomas were snipped and pooled from the same mouse to prepare total RNA. Details on primers and antibodies as well as staining for cell proliferation, cell death and various markers are included in supplemental material (*Supplementary file 3C and D*).

## Quantitative real-time polymerase chain reaction

For gene expression analysis, fresh mucosal scrapings from 10 cm of jejunum were washed in cold PBS, resuspended in 700 µl of RNA lysis buffer, and homogenized in a Dounce homogenizer. RNA was isolated using the Quick-RNA MiniPrep kit (Zymo Research, Orange, CA, USA) according to the manufacturer's instructions. cDNA was generated from 2 to 4 µg of total RNA pooled from three mice using SuperScript III reverse transcriptase (Invitrogen) and random primers. The expression was normalized to the house keeping gene GAPDH. WT values were set 1.

## Tissue processing and staining

The intestines were dissected immediately upon sacrifice, rinsed with cold saline, opened longitudinally and tacked to a foam board for fixation in 10% (vol/vol) formalin. Adenomas were counted under a dissecting microscope, after which tissues were rolled up into 'swiss rolls' for paraffin embedding and histological analysis. Histological analysis was performed by hematoxylin and eosin (H and E) staining.

Immunohistochemistry (IHC) and immunofluorescence (IF). All measures were quantified from swiss rolls of each mouse from three mice per group. Paraffin-embedded sections were subjected to deparaffinization and antigen retrieval (boiling for 10 min in 0.1 M citrate buffer, pH 6.0, with 1 mM EDTA), followed by staining. An Olympus BX51 microscope equipped with SPOT camera and SPOT Advanced 5.1 software was used to acquire the images.

Phospho-eIF4E: Sections were deparaffinized and rehydrated through graded ethanols. Antigen retrieval was performed by boiling for 10 min in 0.1 M citrate buffer (pH 6.0) with 1 mM EDTA. Nonspecific antibody binding was blocked using 20% goat serum (Invitrogen) at room temperature for 30 min. Sections were incubated overnight at 4°C in a humidified chamber with 1:100 diluted rabbit anti-phospho-eIF4E (S209) (76256; Abcam). Sections were then incubated with AlexaFluor 594-conjugated goat anti-rabbit secondary antibodies (1:200; AA11012; Invitrogen) for 1 hr at room temperature (*Yu et al., 2006*). Sections were then washed in PBS and mounted with VectaShield + DAPI (Vector Labs).

Phospho-4E-BP1: Sections were prepared as described above. Non-specific antibody binding was blocked using 20% goat serum (Invitrogen) at room temperature for 30 min. Sections were washed in PBS and incubated overnight at 4°C in a humidified chamber with 1:100 diluted rabbit anti-4E-BP1 (S65/70) (9451; Cell Signaling). Sections were then incubated with AlexaFluor 594- conjugated goat anti-rabbit secondary antibodies (1:200; AA11012; Invitrogen) for 1 hr at room temperature. Sections were then washed in PBS and mounted with VectaShield + DAPI (Vector Labs) for visualization.

Phospho-S6: Sections were prepared as described above. Non-specific antibody binding was blocked using 20% goat serum (Invitrogen) at room temperature for 30 min. Sections were washed in PBS and incubated overnight at 4°C in a humidified chamber with 1:100 diluted rabbit anti-phospho-S6 (S235/236) (2211; Cell Signaling). Sections were then incubated with AlexaFluor 594- conjugated goat anti-rabbit secondary antibodies (1:200; AA11012; Invitrogen) for 1 hr at room temperature. Sections were then washed in PBS and mounted with VectaShield + DAPI (Vector Labs).

Ki67: Sections were prepared as described above. Non-specific antibody binding was blocked using 20% goat serum (Invitrogen) at room temperature for 30 min. Sections were washed in PBS and incubated overnight at 4°C in a humidified chamber with 1:100 diluted rat anti-Ki67 (M7249; DAKO). Sections were then incubated with AlexaFluor 594- conjugated goat anti-rat secondary antibodies (1:200; AA11007; Invitrogen) for 1 hr at room temperature. Sections were then washed in PBS and mounted with VectaShield + DAPI (Vector Labs).

Myc: Sections were prepared as described above. Non-specific antibody binding was blocked using 20% goat serum (Invitrogen) at room temperature for 30 min. Sections were washed in PBS and incubated overnight at 4°C in a humidified chamber with 1:100 diluted rabbit anti-Myc (Abcam 39688). Sections were then incubated with AlexaFluor 594- conjugated goat anti-rabbit secondary antibodies (1:200; AA11012; Invitrogen) for 1 hr at room temperature. Sections were then washed in PBS and mounted with VectaShield + DAPI (Vector Labs).

## Human tissue samples

De-identified Frozen specimens of normal colon and colonic adenomas (*Wang et al., 1998*) were obtained from the University of Pittsburgh Biospecimen CORE with tissue collection under informed consent and usage approved by the Institutional Review Board at the University of Pittsburgh under the protocol of REN11110076/IRB0411047.

Lysates were prepared and used for western blotting as described (*Leibowitz et al., 2014*). In brief, minced pieces of human normal colon and polyp tissue were washed in 1 ml of ice-cold PBS and pelleted at 400 g. Pellets were resuspended in 700 µl of homogenization buffer (0.25 M sucrose, 10 mM Hepes, and 1 mM EGTA) supplemented with protease inhibitors (cOmplete EDTAfree mini, Roche) and homogenized in a Dounce homogenizer with 50 strokes of the pestle. After clearing by centrifugation at 16,000 g, protein concentrations in the supernatant were determined by a spectro-photometer (NanoDrop 2000, Thermo Fisher Scientific). Proteins (20 µg) were separated by the NuPAGE system (Invitrogen) and transferred to polyvinylidene difluoride membranes (Immobilon-P, Millipore). Antibodies used are listed in (*Supplementary file 3D*).

## Human colon organoids

Patient derived CRC organoids were established using surgically resected and de-identified CRC tissues from the Pitt Biospecimen Core (PBC) at University of Pittsburgh. Human colon organoids were developed and cultured as previously described (*Fujii et al., 2015*; *Leibowitz et al., 2018*) and passaged twice a week with a split ratio of 1:3 with minor modifications. In brief, the complete culture medium for human colon organoids contained advanced DMEM/F12 (12634–010; Invitrogen) supplemented with 1x penicillin/streptomycin (15140–122; Invitrogen), 2 mM GlutaMAX (35050–061; Invitrogen), 10 mM HEPES (15630–106; Invitrogen), 1xB27 (17504–044; Invitrogen), 1xN2 (17502–048; Invitrogen), 1 mM N-Acetylcysteine (A0737; Sigma), 10 nM [leu-15]-Gastrin (G9145; Sigma), 10 mM nicotinamide (N0636; Sigma), 10 µM SB202190 (S7067; Sigma), 0.5 µM A83-01(TGFβ inhibitor) (2939; Tocris Bioscience), 20% (vol/vol) FBS (S11150; ATLANTA Biologicals) and WRN-conditioned medium (50%, vol/vol) derived from L-WRN cells (ATCC CRL-3276). Freshly developed organoids were cultured with the complete medium plus 10 µM Y-27632 (Y0503 Sigma) and 100 µg/ml Primo-cin (ant-pm-1, InvivoGen) before the first passage. Key reagents are listed in (*Supplementary file 3F*).

Organoids were feed every 2–3 days with fresh complete medium and passaged approximately 1:3 every week (7–10 days). To passage, organoids in the Matrigel were mechanically disrupted with the growth media by pipetting up and down 10 times. The entire mixture including the pooling of 3–10 wells was transferred to a 15 ml conical tube and centrifuged at 400 x g for 5 min at room temperature. After removal of the top layer of media and second thin layer of Matrigel, organoids at the bottom were digested with 1 ml TryPLE at 37°C for 5 min and centrifuged as above. The pellet was then resuspended in Matrigel 50 µl/well (on ice) and seeding to a new 24-well plate prewarmed to 37°C in the incubator. The plate was put into the incubator for 10 min to solidify the Matrigel followed by addition of 500 µl/well of complete medium. To freeze down, organoids were harvested as above, and resuspended in growth media containing 10% DMSO and 10% FBS, at one well into 1 ml/vial. The vials were stored in liquid nitrogen.

To treat, freshly passaged organoids were seeded into 24-well plates at appropriate density and cultured for 2 days to around 30–50% confluency. The medium was replaced with 500 µl/well of growth medium with or without 10 µM eFT508 (S8275; Selleckchem). After treatment, organoids in triplicate wells of each group were observed and photographed till 48 hr. In addition, organoids from 8 wells of each group were pooled and harvested into100-150 µl protein lysis buffer for western blotting.

## Transcriptomics and bioinformatics

### Microarray analysis

Total RNA was isolated from 2 million cells cultured in T75 flasks with or without treatment for 24 hr. Replica cell pellets were frozen in liquid nitrogen and used later for RNA isolation and RT-PCR analysis for candidate genes. Microarray analysis was performed using Affymetrix Human Genome U133 Plus 2.0 array by Core Facility at the University of Pittsburgh School of Medicine as described (*Wang et al., 2009*).

### Pathway analysis

Differentially expressed genes were identified with fold change (*Vogelstein et al., 2013*; *Dang, 2016*) in various conditions and p-value (equal or less than 0.05). Shared or intersected differentially expressed genes among groups were depicted by Venn Diagram R package (*Chen and Boutros, 2011*). Scatter plot was applied to visualize differentials in selected cell lines or treatment conditions. Log2 of the expression fold change was used as input for the plot.

Reactome pathway analysis (https://reactome.org/) was applied to discover biological functions of the gene set of interest (*Croft et al., 2014*). The input is the genes of interest and the output is the enriched pathways with the significant p-values ($\leq$0.05). The display was limited to top five non overlapping pathways with the most significant p-values.

ClueGO plugin in Cytoscape (http://apps.cytoscape.org/apps/cluego) was additionally used to visualize Reactome Pathway analysis results, and the significance of the terms and groups is automatically calculated with kappa statistics (*Bindea et al., 2009*).

TCGA data mining eIF4E and 4E-BP1 of mRNA and protein expression were analyzed at cBioportal (https://www.cbioportal.org/) using the TCGA provisional cohort (n = 640).

Expression of ISR targets was analyzed at Gene expression Profiling Interactive Analysis (GEPIA) (http://gepia.cancer-pku.cn/). Two cohorts were used, colon adenocarcinoma (COAD, T, n = 275) and rectum adenocarcinoma (READ, T, n = 92) with matched TCGA normal and GTEx data (N, n = 349).

### Data deposit

Gene expression data were generated via cDNA microarray using Affymetrix Human Genome U133 Plus 2.0 array. Microarray data have been deposited at DRYAD (https://dx.doi.org/10.5061/dryad.tb2rbnzxm).

## Statistical analysis

Statistical analyses were carried out using GraphPad Prism software (VIII, GraphPad Software, Inc, La Jolla, CA). Multiple comparisons were analyzed by one-way analysis of variance (ANOVA) followed by Tukey's post-hoc test, whereas those between two groups were made by two-tailed, unpaired *t* test. Differences were considered significant if the probability of the difference occurring by chance was less than 5 in 100 ($p < 0.05$). The means $\pm$ one standard deviation (s.d.) were displayed in the figures. Sample size was determined using a combination of published work and power calculations. For ANOVA, we have computed the power for a test of interaction in a two-way factorial design applied by constructing mixed linear growth models to calculate the needed sample size. We estimated that usually 5 to 10 per group will provide 80% power to detect a standardized interaction of 1.5 SDs, and fewer mice with bigger difference (*Leibowitz et al., 2018*).

## Acknowledgements

Funding: This work is supported in part by NIH grant R01CA215481 and institutional funds (JY), National Key R and D Program of China (2018YFC1313400) (XX), NIH grants R01LM012011 (XL), and R01CA172136 and R01CA203028(LZ). This project used the UPMC Hillman Cancer Center shared glassware, animal, and cell and tissue imaging facilities that were supported, in part, by National Cancer Institute award P30CA047904.

# Additional information

## Competing interests

Nahum Sonenberg: Reviewing editor, *eLife*. The other authors declare that no competing interests exist.

## Funding

| Funder | Grant reference number | Author |
|---|---|---|
| NIH | R01CA215481 | Jian Yu |
| NIH | R01CA172136 | Lin Zhang |
| NIH | R01CA203028 | Lin Zhang |
| NIH | R01LM012011 | Xinghua Lu |
| National Key R and D Program of China | 2018YFC1313400 | Xiang Xu |
| UPMC HCC | Institutional funds | Jian Yu |

The funders had no role in study design, data collection and interpretation, or the decision to submit the work for publication.

## Author contributions

Hang Ruan, Data curation, Formal analysis, Validation, Investigation, Visualization, Methodology, Writing - original draft, Writing - review and editing; Xiangyun Li, Data curation, Formal analysis, Investigation, Methodology, Writing - original draft, Writing - review and editing; Xiang Xu, Resources, Data curation, Formal analysis, Supervision, Funding acquisition, Methodology, Writing - original draft; Brian J Leibowitz, Data curation, Formal analysis, Investigation, Methodology, Writing - review and editing; Jingshan Tong, Methodology; Lujia Chen, Software, Formal analysis, Visualization, Writing - original draft; Luoquan Ao, Wei Xing, Data curation, Investigation, Methodology; Jianhua Luo, Yanping Yu, Resources, Data curation; Robert E Schoen, Resources, Methodology; Nahum Sonenberg, Methodology, Writing - original draft; Xinghua Lu, Resources, Software, Supervision, Funding acquisition, Methodology; Lin Zhang, Resources, Supervision, Funding acquisition, Methodology; Jian Yu, Conceptualization, Resources, Formal analysis, Supervision, Funding acquisition, Writing - original draft, Project administration, Writing - review and editing

## Author ORCIDs

Nahum Sonenberg (iD) http://orcid.org/0000-0002-4707-8759
Xinghua Lu (iD) http://orcid.org/0000-0002-8599-2269
Jian Yu (iD) https://orcid.org/0000-0002-4021-1000

## Ethics

Animal experimentation: This study was performed in strict accordance with the recommendations in the Guide for the Care and Use of Laboratory Animals of the National Institutes of Health. All of the animals were handled according to approved institutional animal care and use committee (IACUC) protocols (# 19085635 and 18063020) of the University of Pittsburgh. All animal experiments were approved by the University of Pittsburgh Institutional Animal Care and Use Committee under Animal welfare assurance number A-3187-01. No surgery or invasive procedure was performed and every effort was made to minimize suffering with humane sacrifice.

## Decision letter and Author response

Decision letter https://doi.org/10.7554/eLife.60151.sa1
Author response https://doi.org/10.7554/eLife.60151.sa2

# Additional files

## Supplementary files

- Supplementary file 1. Differential and selected genes in WT and 4EKI cells.
- Supplementary file 2. Driver information in cell lines used in study.
- Supplementary file 3. Key reagents used in study.
- Transparent reporting form

## Data availability

Microarray data is deposited in Dryad under https://doi.org/10.5061/dryad.tb2rbnzxm. All data generated or analyzed during this study are included in the manuscript and supporting files.

The following dataset was generated:

| Author(s) | Year | Dataset title | Dataset URL | Database and Identifier |
|-----------|------|---------------|-------------|-------------------------|
| Yu J | 2020 | Microarray data | https://dx.doi.org/10.5061/dryad.tb2rbnzxm | Dryad Digital Repository, 10.5061/dryad.tb2rbnzxm |

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
