## [Decision Letter]

**Acceptance summary:**

This study uncovers a novel mechanism by which Myc is up-regulated in colorectal cancers to drive glutamine addiction of these tumors. The data are novel, compelling and clinically-relevant.

**Decision letter after peer review:**

Thank you for submitting your article "eIF4E S209 phosphorylation licenses Myc-and stress-driven oncogenesis" for consideration by *eLife*. Your article has been reviewed by three peer reviewers, one of whom is a member of our Board of Reviewing Editors, and the evaluation has been overseen by Maureen Murphy as the Senior Editor. The following individual involved in review of your submission has agreed to reveal their identity: Zhenghe J Wang (Reviewer #2).

The reviewers have discussed the reviews with one another and the Reviewing Editor has drafted this decision to help you prepare a revised submission.

Summary:

Overall two of three reviewers found the study to be outstanding with high quality data and convincing conclusions. These two reviewers recommended essentially minor revisions that involve some clarifications and additional discussion material to elaborate further on the study's limitations and future directions. The second reviewer found a number of additional points that require clarification and discussion. Please consider all of the reviewers' comments in revising your manuscript and in your point-by-point response. If you have any further data that address points raised, please include them. If you need extra time to repeat any experiments as suggested please do so to improve the quality of the manuscript.

*Reviewer #1:*

eIF4E regulates translation initiation and S209 is often phosphorylated in cancers with deregulated RAS-MAPK or PI3k-mTOR signaling. The authors generated eIF4E^S209A^ knock-in (4EKI) in HCT116 (heterozygous knock-in) and mice (heterozygous knock-in, and 4EKI/KI). They report with 4EKI (S209A/+) in HCT116 reduced Myc and ATF4 translation, inhibition of ISR-dependent glutamine signature, Akt activation and tumor growth. In the min mouse background, 4EKI/KI inhibited polyposis, Myc and Akt activation. They further uncover an interaction between p-eIF4E, Myc and KRAS leading to glutamine addiction that may be a therapeutic vulnerability upon deprivation. The paper unravels important signaling mechanisms involving protein translation reprogramming that are relevant to cellular transformation and which may be exploited therapeutically in cancer.

Overall the paper is well-written and adds important information for a poorly understood cancer progression mechanism involving Myc, Ras, ATF4, p-4E, glutamine, and regulation of protein translation. The data is solid including multiple in vivo models.

The model of S209A results in blockade of phosphorylation of eIF4E. A phosphomimic mutant (S209D or S209E) would have been of interest even just in HCT116 with overexpression and could have provided additional important support for the author's findings.

Please elaborate more in the Discussion on the potential significance of MMP7. The authors could also explain better the underlying mechanism of ISR-dependent glutamine addiction and molecular basis for therapeutic vulnerability. This includes more discussion of how KRAS cooperates with Myc to feed into the glutamine addiction. It should be more clear in the paper that the underlying mechanism for the cooperativity has not been dissected, and neither have the reasons for heterogeneity among KRAS mutant colorectal cancer cell lines. As a start, how does KRAS knockdown or ablation impact on the results in Figure 6 in the different cell lines, including the impact on glutamine deprivation-mediated cell death phenotype.

Abstract: should say glutamine addiction (not glutamine addition)?

Figure 7F should be more clear regarding outcomes of tumor growth, cell death or survival, etc

*Reviewer #2:*

Xuan et al. report here that eIF4E S209 phosphorylation drives high levels of Myc and ATF4 translation, thereby rendering CRC cells dependent on glutamine and promoting tumor growth. Moreover, they further showed that mutant KRAS makes CRC dependent on glutamine through this mechanism. This study uncovers a novel mechanism by which Myc is up-regulated in CRCs to drive glutamine addiction of CRCs. The study is very comprehensive and convincing. Data presented are of high quality. However, the following concerns need to be addressed:

1) They generated two eIF4E S209 mutant knockin clones. In Figure 3, they only performed xenograft study of one knockin clone. To enhance the scientific rigor, they should grow the other knockin clone to show that it also forms small tumors.

2) Figure 6 shows that mutant KRAS cells express high levels of pS209 eIF4E. However, it is not clear what is the molecular mechanism by which mutant KRAS up-regulates pS209 eIF4E.

3) Throughout the manuscript, eIF4E S209 is spelled as eIF4ES209, which may cause confusion to readers outside of the field.

4) In the sentence "Expression of Gln metabolic and translational targets (60-90%) was markedly reduced in 4EKI tumors (Figure 3J)," I believe that they should refer Figure 3K.

5) They state in the sentence "Cell death (TUNEL), p-S6, or expression in ATF4, CHOP, Myc, 4E or apoptotic targets was not significantly different between WT and 4EKI tumors (Figure 3K and Figure 3—figure supplement 1H-I)." However, Figure 3K shows that mRNA levels of ATF4, CHOP and c-MYC are significantly reduced in the knockin clone.

6) The transition to mutant KRAS studies is a bit abrupt. The authors should come up with a better transition.

7) Statistical analyses are missing for some of the quantitative data, e.g., Figure 5 C, G, H and I, and Figure 6C.

8) I assume Figure 5E shown polysome transcripts in WT and KI cells 4 hour and 24 hour after glutamine deprivation and normalized to WT with Gln. If so, please clarify in the figure legend.

9) To be clear, Figure 5F should be labeled as total RNA instead of T-RNA.

10) The schematics shown in Figure 7F does not clearly express the point that glutamine deprivation down-regulates c-MYC. The authors should modify it to make the point come across clearly. They should integrate mutant KRAS into the diagram as well.

*Reviewer #3:*

The manuscript by Ruan and colleagues addresses the yet unresolved important question of the role of eIF4E S209 phosphorylation in colorectal tumorigenesis. Using non-phosphorylatable mutant (eIF4E S209A) knockin CRC cells and mouse models, they uncover a mechanism of p-eIF4E inducing ISR-dependent MYC and ATF4 translation leading to changes in glutamine metabolism and oncogenic proliferation. They suggest an oncogenic function of p-eIF4E S209 in CRC initiation and progression. Although the manuscript provides new data and insight on the importance of eIF4E phosphorylation for CRC tumorigenesis, which is of relevance as potential therapeutic target, there are substantial discrepancies between the data and the proposed links to stress signaling, cell death and immune evasion. Furthermore, parts of the data linking metabolic stress by glutamine deprivation to p-4E and cell death are opposing to already published mechanisms, and this is not discussed.

1) Subsection “eIF4E S209 regulates colon cancer cell growth and Myc translation”, Figure 1A. Only later it is shown that AKT activity is reduced, so signaling proteins upstream of 4EBP1 should be analysed also here in both clones for their phosphorylation status. Furthermore, 4EBP T37/46 phosphorylation is known as priming event for S65/70 phosphorylation meaning it would be counterintuitive that the latter is not affected.

2) Although introducing 4E S209 mutation is an elegant system to explore its role in tumorigenesis, neither homozygous 4EKI HCT116 clones could be obtained nor heterozygous RKO or HT29 4EKI clones, as explained by the authors. Expansion of data to more cell lines would have been important, of course, especially as RKO and HT29 are also used in later experiments. Likewise, as 4E is only heterozygously mutated, observed affects are hard to control. The authors could use a more physiological system as intestinal organoids to extend and support their findings. Furthermore, the kinases MNK1 and MNK2 are the only characterized kinases that phosphorylate 4E S209. As there are efficient MNK1/2 inhibitors available, which can also be used in vivo, the authors could use those to validate their most important findings.

3) To analyse changes in global translation efficiency in 4EKI cells the authors used polysome profiling (Figure 2A) and luciferase reporter assays (Figure 2B,C, Figure 2—figure supplement 1). It looks like that the whole polysome profile of 4EKI cells is shifted down in comparison to the profile of wt cells making interpretation difficult. In addition, using exogenously expressed luciferase reporters is a very artificial system to study endogenous protein synthesis. It is recommended to analyse global protein synthesis via S35-Methionine incorporation or Puromycin labelling, especially as observed translation changes are very subtle. Furthermore, if bicistronic reporter vectors are used to distinguish between cap- and IRES-dependent translation, a hairpin structure should be introduced in front of the IRES sequence. This can inhibit potential read-through of ribosomes, which can affect interpretation of results.

4) Figure 2E-H. The authors show reduced translation of MYC/MMP7 in 4EKI cells. Although polysomal mRNA is indeed more reduced than total RNA (at least shown for clone 1), also transcription is affected. Ribosomal profiling experiments would help to clearly distinguish between transcriptional and translational effects and to analyse whether a global oncogenic translational program is regulated by 4E S209 phosphorylation.

5) Subsection “eIF4E S209 promotes colon cancer growth in vivo through Myc and the ISR”, Figure 3D. The authors performed siRNA-mediated knockdown of MYC/ATF4/CHOP and show downregulation of Gln genes. It is not obvious, what is the claim behind that experiment. That knockdown of those proteins has similar effects as observed in 4EKI cells that show decreased expression of Gln genes and, therefore, these effects are MYC-/stress-dependent? This link is not directly shown by this experimental setting. Instead, the effects in 4EKI vs. wt cells should be analyzed upon MYC/ATF4 overexpression (as performed in Figure 5H and Figure 5—figure supplement 1B) and whether downregulation of Gln genes can be rescued by this.

6) Figure 3G. The immunofluorescence staining is not convincing, especially p-GCN2/p-eIF2 stainings that only show some single spots rather than broad cytoplasmic staining. A positive control should be used to validate a specific signal (e.g. tunicamycin treatment for p-eIF2a). Also, stainings of total GCN2/eIF2a/AKT levels should be performed. Please provide H&E stainings.

7) The authors state "Cell death (TUNEL), p-S6, or expression in ATF4, CHOP, Myc, 4E or apoptotic targets was not significantly different between WT and 4EKI tumors (Figure 3K and Figure 3—figure supplement 1H-I). The expression of ISR Gln metabolic and apoptotic targets (ASCT2, SLC7A5, PUMA, TRIB, and NOXA) is also elevated in TCGA CRC cohorts".

This description is not correct, as transcript levels of ATF4/CHOP/MYC (Figure 3K) and also some apoptotic genes (DR5, NOXA Figure 3—figure supplement 1I) are obviously significantly downregulated in 4EKI vs. wt tumors. The description and relevance of apoptosis between wt and 4EKI samples is not clear/confusing, especially as highly significant downregulation of apoptotic genes in 4EKI vs. wt cells is not described at all (Figure3—figure supplement 1F).

Please adjust description and explain in more detail.

8) The authors conclude "These data support that p-4E promotes Myc- and ATF4-driven CRC growth through chronic exploitation of the most ancient arm of ISR (GCN2) to maintain constitutive AKT/4E-BP1 signaling."

The conclusion that p-4E promotes the ISR and AKT/4EBP signaling via p-eIF2a/p-pGCN2 is mainly based on the not convincing tumor stainings (Figure 3G), see also comment 6. How do levels of p-eIF2a/pGCN2/pAKT levels react in 4EKI vs. wt HCT116 cells? Immunoblots should be shown. The authors could induce GCN2 activation (e.g. amino acid starvation, RNA Pol III inhibitor) to test whether active GCN2 rescues the phenotype in 4EKI cells/tumors. Furthermore, the authors argue that this MYC/ATF4/ISR-driven AKT/4EBP signaling fuels enhanced protein synthesis and, subsequently, tumor growth. Although enhanced protein synthesis is an essential factor for MYC-driven tumorigenesis, several recent publications have shown that this needs to be tightly balanced by several mechanisms. It would help for understanding, if the author's findings would be discussed more in the context of those publications.

9) Figure 4A/Figure 4—figure supplement 1A. The authors state “Uniform p-4E staining was detected in the bottom of proliferating (Ki-67+) crypts in normal mice". There is no overlay or co-staining (p-4E and Ki67) from the same section shown, making it impossible to interpret whether p-4E is found in Ki67+ crypts. Additionally, higher magnification of p-4E staining should be shown (as in Figure 4—figure supplement 1A). Please provide H&E stainings.

10) Figure 4D/E. The authors state "Compared to polyps formed in the control mice at similar sizes, ones formed in 4EKI mice showed strongly reduced levels of c-Myc, p-AKT, p-4E-BP1". This raises the question: If tumor formation is driven by elevated p-4E/MYC/pAKT/p-4EBP levels, why do those polyps form despite reduced levels of those proteins? Are those polyps smaller, although it does not look like in Figure 4B? How do p-4E levels in these APC^min^ 4EKI/KI tumors behave, stainings should be shown. It is recommended to show all stainings for total proteins (4EBP/eIF2a/AKT).

11) The authors state “Both glucose and Gln were required for the growth of HCT116 cells in culture… (Figure 5A-C)." This metabolic link has been published before, specifically in HCT116 wt cells, please cite properly (Dejure et al., 2017). In this previous publication, a mechanism has been well established that Gln starvation induces only moderate apoptosis but rather S phase arrest which is dependent on the MYC 3'UTR. Please comment on this. The authors should include BrdU π cell cycle FACS analysis to evaluate these contradictory results.

12) Please include p values for FACS analyses in complete Figure 5 and Figure 5—figure supplement 1.

13) The authors state “Apoptotic targets (DR5, PUMA and NOXA) showed transient suppression of translation at 4 hours followed by strong increase by 24 hours (Figure 5D, middle)." NOXA is clearly downregulated after 24h, please correct.

14) The authors state “In contrast, 4EKI cells showed reduced basal levels of Myc, ISR and p-AKT, and transient ISR induction… The kinetics or levels of p-4E-BP1 (T37/46), p-ERK, p-mTOR (S2448), p-S6, p-PERK, cyclin D1 were not significantly affected by 4EKI (Figure 5D)". Although I appreciate that p-eIF2a basal levels are lower in 4EKI cells, the other ISR markers (ATF4 and CHOP) seem to be not expressed at all in Figure 5D (middle panel) in both wt and 4EKI, so the claim that ISR basal levels are reduced in 4EKI cells is questionable. Also, it does not fit with Figure 3B where it is shown that ATF4 and CHOP total and polysomal mRNA levels are reduced in 4EKI vs. wt cells. Additionally, quantification of immunoblots, especially right panel, would make it easier to judge changes in kinetics of pmTOR/ERK/S6/AKT. Changes in p-PERK levels upon Gln deprivation are clearly not the same (more stable in wt than in 4EKI), in contrast to what is stated by the authors. Loading control is missing in Figure 5D, middle panel.

15) Subsection “eIF4E S209 promotes Myc- and ISR-dependent glutamine addiction”. Figure 5E-H are displayed in a confusing way and important controls are missing. Figure 5E needs to be shown for total RNA in the same way, otherwise described kinetics in changes of gene expression are not comprehensible (-GLN only shown for 24h for total RNA in F). MYC is not shown at all in total RNA, although it is stated that "Myc loss and ATF4 induction was rapid and primarily in polysomal fractions". What is lane labelled with "GLN+" in Figure 5F, wt cells? If yes, 4EKI cells GLN+ also have to be shown as comparison as zero time points. This discrepancy would be avoided if same heatmap of total RNA is shown.

siRNA experiments and MYC/ATF4 overexpression experiments in Figure 5F/G/H and Figure 5—figure supplement 1A/B should be shown in both wt and 4EKI cells side-by-side. In Figure 5G, the authors claim that siCHOP/ASCT2/ASNS reduces apoptosis to a lesser extent upon GLN deprivation. Nevertheless, the fold induction of apoptosis (GLN+ vs. GLN-) seems to be similar to siCTR cells. Same is true for Figure 5H where fold induction of apoptosis looks similar between CTR and MYC overexpressing cells. ATF4 immunoblot is missing in Figure 5—figure supplement 1B.

16) Subsection “Mutant KRAS cooperates with p-4E/Myc to promote glutamine addiction and immune suppression”, Figure 6A. Total eIF4E levels are also elevated in Lim1215 cells with KRAS mutations, arguing against increased p-4E levels in this cell line. Quantification of immunoblot could help clarifying this.

17) Subsection “Mutant KRAS cooperates with p-4E/Myc to promote glutamine addiction and immune suppression”, Figure 6C,D. The apoptotic response in HCT116 wt cells is much lower than in previous assays in HCT116 (26% in Figure 6B vs. over 50% in Figure 5) and no apoptosis induction at all in Lim1215 wt cells upon GLN deprivation (also no or only weak induction of ISR, and no cl. Casp 3, Figure 6D). This raises the question to which extent this phenotype/mechanism can be generalized to CRC or whether the mutational landscape of different CRC cells affect the response. The number of analysed cell lines could be extended or a more physiological, genetically defined system with specific mutations such as intestinal organoids could be used.

18) The authors state “The sensitivity was associated with elevated apoptosis and p-GCN2/ISR response, and reduction in Myc but p-ERK (Figure 6C-E, Figure 6—figure supplement 1A-C)." MYC levels are not shown in any of the given figures. Why should MYC levels be reduced in KRASmut cells upon GLN deprivation if they are more sensitive to deprivation and if this is mediated via MYC induction?

19) The authors state “These data establish that p-4E/Myc cooperates with mutant KRAS to promote glutamine addiction with elevated ISR and immune suppression". Are transcriptome analyses in Figure 6F-J performed in KRASwt cells? This is not completely obvious from the description. If yes, then the claim that cooperation with KRAS promotes immune suppression is not reflected by the data. Global gene expression changes are then only shown for KRASwt cells, same analyses need to be performed in KRASmut cells (-/+GLN, wt/4EKI).

20) Figure 7B. Total eIF4E and 4EBP levels are also elevated in RKO and HT29 cells, arguing against increased p-4E/p-4EBP levels. Quantification of immunoblot would help clarifying this.

21) The authors state “Together, our data support that p-4E controls cell death upon metabolic stress". This conclusion is only partially reflected by the data. I appreciate that HCT116 cells show reduced p-4E levels upon metabolic stress (GLN deprivation) associated with enhanced stress signaling (Figure 5). Instead, neither HT29 nor RKO cells that are sensitive to GLN deprivation with enhanced stress signaling, show changes in p-4E levels (Figure 7B). Therefore, it is questionable whether the proposed mechanism, p-4E regulating the response to metabolic stress, is a general feature of CRC. On the same line, a clear mechanism of the response to GLN deprivation in CRC cells has been described previously, that is associated with reduced nucleotide and MYC levels and, subsequently, slowed elongation of RNA Pol II and cell cycle arrest instead of apoptosis. This is not at all discussed in the manuscript.

22) The depicted model in Figure 7F is not comprehensible.

---

## [Author Response]

Reviewer #1:eIF4E regulates translation initiation and S209 is often phosphorylated in cancers with deregulated RAS-MAPK or PI3k-mTOR signaling. The authors generated eIF4E^S209A^ knock-in (4EKI) in HCT116 (heterozygous knock-in) and mice (heterozygous knock-in, and 4EKI/KI). They report with 4EKI (S209A/+) in HCT116 reduced Myc and ATF4 translation, inhibition of ISR-dependent glutamine signature, Akt activation and tumor growth. In the min mouse background, 4EKI/KI inhibited polyposis, Myc and Akt activation. They further uncover an interaction between p-eIF4E, Myc and KRAS leading to glutamine addiction that may be a therapeutic vulnerability upon deprivation. The paper unravels important signaling mechanisms involving protein translation reprogramming that are relevant to cellular transformation and which may be exploited therapeutically in cancer.Overall the paper is well-written and adds important information for a poorly understood cancer progression mechanism involving Myc, Ras, ATF4, p-4E, glutamine, and regulation of protein translation. The data is solid including multiple in vivo models.The model of S209A results in blockade of phosphorylation of eIF4E. A phosphomimic mutant (S209D or S209E) would have been of interest even just in HCT116 with overexpression and could have provided additional important support for the author's findings.

p-4E (and mTOR/AKT) is highly elevated in CRC, we envision that it would be difficult to predict or explain the effects of S209D (E) in CRC. The loss of (hypo) function mutant (hypo phosphorylation) is therefore more informative on dissect its role in Myc and ISR-dependent CRC growth. We discussed the punctate signals and dynamic and local nature of GCN2-eIF2a and 4E-4EBP1 interaction. See Results (subsection “eIF4E S209 promotes colon cancer growth in vivo through Myc and the ISR”) and Discussion.

Please elaborate more in the Discussion on the potential significance of MMP7. The authors could also explain better the underlying mechanism of ISR-dependent glutamine addiction and molecular basis for therapeutic vulnerability. This includes more discussion of how KRAS cooperates with Myc to feed into the glutamine addiction. It should be more clear in the paper that the underlying mechanism for the cooperativity has not been dissected, and neither have the reasons for heterogeneity among KRAS mutant colorectal cancer cell lines. As a start, how does KRAS knockdown or ablation impact on the results in Figure 6 in the different cell lines, including the impact on glutamine deprivation-mediated cell death phenotype.

Our data suggest that MMP7 expression is regulated by transcription and translation in p-4E high cells, much like other ISR effectors. We elaborated the rationale with background on mutant *KRAS/BRAF*, Mnk1/2 and elevated p-4E, as well p-4E-dependent translation of other oncogenic targets (i.e. PD-L1, SNAIL and MMP-3). Our new data showed that a selective Mnk inhibitor reduces the growth of CRC organoids along with reduced p-4E/p-4E-BP1 and Myc. See new Figure 2I-J, revised Abstract, Introduction, Results and Discussion.

Abstract: should say glutamine addiction (not glutamine addition)?

Corrected.

Figure 7F should be more clear regarding outcomes of tumor growth, cell death or survival, etc

We simplified the model to propose the p-4E drives Myc and Gln-dependent metabolic adaption, which is disrupted by Gln withdraw, leading to Myc reduction and ISR hyperactivation. Nutrient- and p-4E sensitive ISR therefore dynamically regulates CRC outcomes during evolution and therapy. See revised Figure 7F and Discussion.

Reviewer #2:Xuan et al. report here that eIF4E S209 phosphorylation drives high levels of Myc and ATF4 translation, thereby rendering CRC cells dependent on glutamine and promoting tumor growth. Moreover, they further showed that mutant KRAS makes CRC dependent on glutamine through this mechanism. This study uncovers a novel mechanism by which Myc is up-regulated in CRCs to drive glutamine addiction of CRCs. The study is very comprehensive and convincing. Data presented are of high quality. However, the following concerns need to be addressed:1) They generated two eIF4E S209 mutant knockin clones. In Figure 3, they only performed xenograft study of one knockin clone. To enhance the scientific rigor, they should grow the other knockin clone to show that it also forms small tumors.

We performed new experiments. KI2 cells showed even more reduced engraftment and growth rate compared to WT cells (on the opposite flank of same mice). See revised Figure 3E and Results section.

2) Figure 6 shows that mutant KRAS cells express high levels of pS209 eIF4E. However, it is not clear what is the molecular mechanism by which mutant KRAS up-regulates pS209 eIF4E.

See reviewer 1 (point 2). We elaborated the rationale on mutant *KRAS/BRAF*, Mnk1/2 and increased p-4E. Our new data showed that a selective Mnk inhibitor reduces the growth of CRC organoids, p-4E/p-4E-BP1 and Myc. See new Figure 2I-J, Revised model Figure 7F, Abstract, Introduction, Results and Discussion sections.

3) Throughout the manuscript, eIF4E S209 is spelled as eIF4ES209, which may cause confusion to readers outside of the field.

We now use eIF4E S209 throughout the manuscript.

4) In the sentence "Expression of Gln metabolic and translational targets (60-90%) was markedly reduced in 4EKI tumors (Figure 3J)," I believe that they should refer Figure 3K.

Corrected.

5) They state in the sentence "Cell death (TUNEL), p-S6, or expression in ATF4, CHOP, Myc, 4E or apoptotic targets was not significantly different between WT and 4EKI tumors (Figure 3K and Figure 3—figure supplement 1H-I)." However, Figure 3K shows that mRNA levels of ATF4, CHOP and c-MYC are significantly reduced in the knockin clone.

Corrected in the revised Results.

6) The transition to mutant KRAS studies is a bit abrupt. The authors should come up with a better transition.

Please see point 2.

7) Statistical analyses are missing for some of the quantitative data, e.g., Figure 5 C, G, H and I, and Figure 6C.

Flow cytometry was repeated on two or more different occasions (days) with similar results. Plots and quantitation from one representative experiment are shown with fraction (%) of given population per convention. Revised MandM and Legends.

8) I assume Figure 5E shown polysome transcripts in WT and KI cells 4 hour and 24 hour after glutamine deprivation and normalized to WT with Gln. If so, please clarify in the figure legend.

Yes. The value of WT with Gln was set as 1 or control for each maker. Revised Legends.

9) To be clear, Figure 5F should be labeled as total RNA instead of T-RNA.

See above. We used Total-RNA to replace T-RNA in any figure as applicable. Values were normalized to WT cell with Gln (+) as one similarly as 5E. *Myc* was added to total-RNA panel. See revised Figure 5F and Results section.

10) The schematics shown in Figure 7F does not clearly express the point that glutamine deprivation down-regulates c-MYC. The authors should modify it to make the point come across clearly. They should integrate mutant KRAS into the diagram as well.

Please see reviewer 1 (point 4). Simplified model with drivers (KRAS), p-4E/p-4E-BP1, Myc/ISR and outcomes. See revised Figure 7F.

Reviewer #3:The manuscript by Ruan and colleagues addresses the yet unresolved important question of the role of eIF4E S209 phosphorylation in colorectal tumorigenesis. Using non-phosphorylatable mutant (eIF4E S209A) knockin CRC cells and mouse models, they uncover a mechanism of p-eIF4E inducing ISR-dependent MYC and ATF4 translation leading to changes in glutamine metabolism and oncogenic proliferation. They suggest an oncogenic function of p-eIF4E S209 in CRC initiation and progression. Although the manuscript provides new data and insight on the importance of eIF4E phosphorylation for CRC tumorigenesis, which is of relevance as potential therapeutic target, there are substantial discrepancies between the data and the proposed links to stress signaling, cell death and immune evasion. Furthermore, parts of the data linking metabolic stress by glutamine deprivation to p-4E and cell death are opposing to already published mechanisms, and this is not discussed.

Thank you for commenting on the significance of our work. Our data supports a critical role of p-4E in coordinating Myc and mutant *KRAS* in control CRC growth, death and transcriptional heterogeneity depending on the nutrient availability (i.e. Gln). These oncogenic properties as well as “Stress” are strongly reduced in 4EKI (KI/+) CRC cells containing a complex mix of driver mutations such *KRAS*, *b-Catenin*, *PIK3CA*, *TGFBR2.*

Importantly, ISR-dependent transcriptional heterogeneity and outcomes (arrest vs cell death) is consistent with our model through Gln and Myc-dependent recovery vs. crisis. Revised Figure 7F, Results and Discussion as detailed below.

1) Subsection “eIF4E S209 regulates colon cancer cell growth and Myc translation”, Figure 1A. Only later it is shown that AKT activity is reduced, so signaling proteins upstream of 4EBP1 should be analysed also here in both clones for their phosphorylation status. Furthermore, 4EBP T37/46 phosphorylation is known as priming event for S65/70 phosphorylation meaning it would be counterintuitive that the latter is not affected.

Added p-AKT, p-mTOR and p-ERK. See revised Figure 1A and Results section.

2) Although introducing 4E S209 mutation is an elegant system to explore its role in tumorigenesis, neither homozygous 4EKI HCT116 clones could be obtained nor heterozygous RKO or HT29 4EKI clones, as explained by the authors. Expansion of data to more cell lines would have been important, of course, especially as RKO and HT29 are also used in later experiments. Likewise, as 4E is only heterozygously mutated, observed affects are hard to control. The authors could use a more physiological system as intestinal organoids to extend and support their findings. Furthermore, the kinases MNK1 and MNK2 are the only characterized kinases that phosphorylate 4E S209. As there are efficient MNK1/2 inhibitors available, which can also be used in vivo, the authors could use those to validate their most important findings.

See reviewer 1 (point 2). We elaborated on mutant KRAS/BRAF, Mnk, and added new data on Mnki in CRC organoids. See new Figure 2I-J, revised Introduction and Results section.

3) To analyse changes in global translation efficiency in 4EKI cells the authors used polysome profiling (Figure 2A) and luciferase reporter assays (Figure 2B,C, Figure 2—figure supplement 1). It looks like that the whole polysome profile of 4EKI cells is shifted down in comparison to the profile of wt cells making interpretation difficult. In addition, using exogenously expressed luciferase reporters is a very artificial system to study endogenous protein synthesis. It is recommended to analyse global protein synthesis via S35-Methionine incorporation or Puromycin labelling, especially as observed translation changes are very subtle. Furthermore, if bicistronic reporter vectors are used to distinguish between cap- and IRES-dependent translation, a hairpin structure should be introduced in front of the IRES sequence. This can inhibit potential read-through of ribosomes, which can affect interpretation of results.

This is a valid point worth studying. However, we chose to focus on oncogenic targets such as Myc, MMP7, and ISR-dependent regulation of Gln metabolism, and validation in multiple CRC cell line and mouse models. See below 4 as well. See revised model Figure 7F and Discussion section.

4) Figure 2E-H. The authors show reduced translation of MYC/MMP7 in 4EKI cells. Although polysomal mRNA is indeed more reduced than total RNA (at least shown for clone 1), also transcription is affected. Ribosomal profiling experiments would help to clearly distinguish between transcriptional and translational effects and to analyse whether a global oncogenic translational program is regulated by 4E S209 phosphorylation.

We focused on Myc and ISR targets in the context of Gln-dependent growth in CRC. The implication of heterogeneity, the use of systematic approach and appropriate models have been discussed. See revised Discussion section.

5) Subsection “eIF4E S209 promotes colon cancer growth in vivo through Myc and the ISR”, Figure 3D. The authors performed siRNA-mediated knockdown of MYC/ATF4/CHOP and show downregulation of Gln genes. It is not obvious, what is the claim behind that experiment. That knockdown of those proteins has similar effects as observed in 4EKI cells that show decreased expression of Gln genes and, therefore, these effects are MYC-/stress-dependent? This link is not directly shown by this experimental setting. Instead, the effects in 4EKI vs. wt cells should be analyzed upon MYC/ATF4 overexpression (as performed in Figure 5H and Figure 5—figure supplement 1B) and whether downregulation of Gln genes can be rescued by this.

We agree and clarified this point. p-4E/Myc/ISR(p-eIF2a/ATF4) are upstream events driving glutamine-biased metabolism and addiction. siRNA of downstream effectors (CHOP, ASNS or ASCT2) has more modest effects Gln addiction. This is consistent with prior studies on ISR level-dependent outcomes, including cell death in MEFs with uncontrolled translational load (*1*), or survival in HCT 116 cells with Myc recovery (*2*). See revised Figure 7F and Discussion section.

6) Figure 3G. The immunofluorescence staining is not convincing, especially p-GCN2/p-eIF2 stainings that only show some single spots rather than broad cytoplasmic staining. A positive control should be used to validate a specific signal (e.g. tunicamycin treatment for p-eIF2a). Also, stainings of total GCN2/eIF2a/AKT levels should be performed. Please provide H&E stainings.

We included higher magnification images p-GCN2 and p-eIF2a while keeping the originals with other markers at the same magnification. The punctate p-GCN2 and/or p-eIF2a staining was observed in the polyps of APC^Min^ mice (Figure 4A). These findings are consistent with moderate increase in ISR and transient and localized GCN2-eIF2a regulation during cell growth (*3*). In contrast, acute metabolic stress (i.e., mTORi) or ER stressors induce high levels of p-eIF2a (ISR) and cell death in HCT 116 xenografts as we have shown (*4*). Total GCN2/eIF2a/AKT levels were not affected by acute metabolic stress in various CRC lines examined. See revised Figure 3—figure supplement 1H, Results and Discussion sections.

7) The authors state "Cell death (TUNEL), p-S6, or expression in ATF4, CHOP, Myc, 4E or apoptotic targets was not significantly different between WT and 4EKI tumors (Figure 3K and Figure 3—figure supplement 1H-I). The expression of ISR Gln metabolic and apoptotic targets (ASCT2, SLC7A5, PUMA, TRIB, and NOXA) is also elevated in TCGA CRC cohorts".This description is not correct, as transcript levels of ATF4/CHOP/MYC (Figure 3K) and also some apoptotic genes (DR5, NOXA Figure 3—figure supplement 1I) are obviously significantly downregulated in 4EKI vs. wt tumors. The description and relevance of apoptosis between wt and 4EKI samples is not clear/confusing, especially as highly significant downregulation of apoptotic genes in 4EKI vs. wt cells is not described at all (Figure 3—figure supplement 1F).Please adjust description and explain in more detail.

Sorry for the oversight. More details are added. The expression of metabolic and translational targets is reduced more significantly compared to apoptotic targets in 4EKI xenografts. The proliferation, p-GCN2/p-eIF2a, and p-AKT but not apoptosis, are reduced in 4EKI xenografts, supporting Gln-dependent metabolic adaptation and growth. See revised Results section.

8) The authors conclude "These data support that p-4E promotes Myc- and ATF4-driven CRC growth through chronic exploitation of the most ancient arm of ISR (GCN2) to maintain constitutive AKT/4E-BP1 signaling."The conclusion that p-4E promotes the ISR and AKT/4EBP signaling via p-eIF2a/p-pGCN2 is mainly based on the not convincing tumor stainings (Figure 3G), see also comment 6. How do levels of p-eIF2a/pGCN2/pAKT levels react in 4EKI vs. wt HCT116 cells? Immunoblots should be shown. The authors could induce GCN2 activation (e.g. amino acid starvation, RNA Pol III inhibitor) to test whether active GCN2 rescues the phenotype in 4EKI cells/tumors. Furthermore, the authors argue that this MYC/ATF4/ISR-driven AKT/4EBP signaling fuels enhanced protein synthesis and, subsequently, tumor growth. Although enhanced protein synthesis is an essential factor for MYC-driven tumorigenesis, several recent publications have shown that this needs to be tightly balanced by several mechanisms. It would help for understanding, if the author's findings would be discussed more in the context of those publications.

See point 6 on punctate staining of p-eIF2a/p-GCN2, which is further supported by data from Apc^min^ mice and human adenomas (Figure 4 and Figure 4—figure supplement 1). p-4E-depedent increase in Myc/p-eIF2a/p-AKT is clearly not required for the rapid proliferation of “Normal” crypts but polyposis driven by oncogenic proliferation.

Gln-dependent adaptation in mutant *KRAS*, p-4E/Myc high tumors is the very cause of their Gln addiction and transcriptional heterogeneity (Figures 5-7). The findings are consistent with the degree and extent of ISR in relation to transcriptional heterogeneity and outcomes. See revised Figure 7F and Discussion section.

9) Figure 4A/Figure 4—figure supplement 1A. The authors state “Uniform p-4E staining was detected in the bottom of proliferating (Ki-67+) crypts in normal mice". There is no overlay or co-staining (p-4E and Ki67) from the same section shown, making it impossible to interpret whether p-4E is found in Ki67+ crypts. Additionally, higher magnification of p-4E staining should be shown (as in Figure 4—figure supplement 1A). Please provide H&E stainings.

It is well established (including by us) that intestinal proliferation is confined in the progenitors and the lower half of the (ALL/every) crypt. We added roomed in Ki-67 and p-4E IF in different crypts, and H&E stained polyps from both WT and 4EKI APC^Min^ mice. See revised Figure 4—figure supplement 1C-D and Results section.

10) Figure 4D/E. The authors state "Compared to polyps formed in the control mice at similar sizes, ones formed in 4EKI mice showed strongly reduced levels of c-Myc, p-AKT, p-4E-BP1". This raises the question: If tumor formation is driven by elevated p-4E/MYC/pAKT/p-4EBP levels, why do those polyps form despite reduced levels of those proteins? Are those polyps smaller, although it does not look like in Figure 4B? How do p-4E levels in these APC^min^ 4EKI/KI tumors behave, stainings should be shown. It is recommended to show all stainings for total proteins (4EBP/eIF2a/AKT).

4EKI reduced the incidence of polyps by over 60%, and lost ones can be studied. We selected polyps at similar sizes (large) to control for growth or metabolic pressure. The residual 4EKI tumors are negative for p-4E with reduced Myc (still elevated compared to normal crypts). A potential role of p-4E in non-epithelial compartments or p-4E-independent mechanisms cannot be ruled out. See revised Results section.

11) The authors state “Both glucose and Gln were required for the growth of HCT116 cells in culture… (Figure 5A-C)." This metabolic link has been published before, specifically in HCT116 wt cells, please cite properly (Dejure et al., 2017). In this previous publication, a mechanism has been well established that Gln starvation induces only moderate apoptosis but rather S phase arrest which is dependent on the MYC 3'UTR. Please comment on this. The authors should include BrdU π cell cycle FACS analysis to evaluate these contradictory results.

See point 5. Our data showed dose-dependent induction of cell death with growth inhibition upon Gln starvation in HCT 116 (Figure 5—figure supplement 1C-D), supporting ISR-dependent outcomes. See revised Figure 7F and Discussion section.

12) Please include p values for FACS analyses in complete Figure 5 and Figure 5—figure supplement 1.

Please see reviewer #2 (point 7). See Materials and methods section and figure legends.

13) The authors state “Apoptotic targets (DR5, PUMA and NOXA) showed transient suppression of translation at 4 hours followed by strong increase by 24 hours (Figure 5D, middle)." NOXA is clearly downregulated after 24h, please correct.

Sorry for the oversight. Corrected. See Results section.

14) The authors state “In contrast, 4EKI cells showed reduced basal levels of Myc, ISR and p-AKT, and transient ISR induction… The kinetics or levels of p-4E-BP1 (T37/46), p-ERK, p-mTOR (S2448), p-S6, p-PERK, cyclin D1 were not significantly affected by 4EKI (Figure 5D)". Although I appreciate that p-eIF2a basal levels are lower in 4EKI cells, the other ISR markers (ATF4 and CHOP) seem to be not expressed at all in Figure 5D (middle panel) in both wt and 4EKI, so the claim that ISR basal levels are reduced in 4EKI cells is questionable. Also, it does not fit with Figure 3B where it is shown that ATF4 and CHOP total and polysomal mRNA levels are reduced in 4EKI vs. wt cells. Additionally, quantification of immunoblots, especially right panel, would make it easier to judge changes in kinetics of pmTOR/ERK/S6/AKT. Changes in p-PERK levels upon Gln deprivation are clearly not the same (more stable in wt than in 4EKI), in contrast to what is stated by the authors. Loading control is missing in Figure 5D, middle panel.

The same set of samples are used for the entire sets with action done on two separate blots (after stripping) for Figure 5D. Additional controls include total eIF2a, GCN2 and cyclin D, which did not change in response to Gln- in either WT or KI cells (or other CRCs).

p-eIF2a/ATF4/CHOP as well as ISR targets assessed by RT-PCR and/or micro array support that the basal ISR is lower in 4E KI HCT 116 cells and xenografts (with Gln) and some are punctate. Basal p-ERK, p-S6 and p-mTOR were similar in WT and 4EKI cells. See revised Figure 1A and Results section.

15) Subsection “eIF4E S209 promotes Myc- and ISR-dependent glutamine addiction”. Figure 5E-H are displayed in a confusing way and important controls are missing. Figure 5E needs to be shown for total RNA in the same way, otherwise described kinetics in changes of gene expression are not comprehensible (-GLN only shown for 24h for total RNA in F). MYC is not shown at all in total RNA, although it is stated that "Myc loss and ATF4 induction was rapid and primarily in polysomal fractions". What is lane labelled with "GLN+" in Figure 5F, wt cells? If yes, 4EKI cells GLN+ also have to be shown as comparison as zero time points. This discrepancy would be avoided if same heatmap of total RNA is shown.siRNA experiments and MYC/ATF4 overexpression experiments in Figure 5F/G/H and Figure 5—figure supplement 1A/B should be shown in both wt and 4EKI cells side-by-side. In Figure 5G, the authors claim that siCHOP/ASCT2/ASNS reduces apoptosis to a lesser extent upon GLN deprivation. Nevertheless, the fold induction of apoptosis (GLN+ vs. GLN-) seems to be similar to siCTR cells. Same is true for Figure 5H where fold induction of apoptosis looks similar between CTR and MYC overexpressing cells. ATF4 immunoblot is missing in Figure 5—figure supplement 1B.

Figures 5E-I and Figure 5—figure supplement 1 examined the role of Myc/ATF4/CHOP in basal and induction of ISR upon Gln- using loss of function and gain of function in WT and 4E KI cells, respectively. These experiments support that the (inducible) expression of ISR targets and death upon persistent ISR, expression of apoptotic targets and failed recovery. Upstream regulators such as Myc (loss), ATF4 and CHOP are (more rapidly) controlled by translation (within 4 hours) (Figure 3A-D, Figure 5D).

5E-F,we clarified in legends on qRT-PCR normalization. Each marker is normalized to parental HCT 116 cells with Gln (1, black) in either sets (reviewer 2/point 8). Revised Figure 5F.

5H. Overexpression of Myc induces apoptosis in p53 WT cells (*5*), and caused a further increase with Gln- in 4EKI cells. Myc siRNA reduced cell death with Gln+ and Gln- in WT cells (Figure 5G). ATF4 overexpression did not increase cell death with Gln while strongly increased cell death upon Gln-. We’d like to argue that absolute levels of ISR effectors are important for cell fate. For example, expression from 0.2 to 1 vs 2-10 (same 5-fold) is likely to cause a different outcome. By analogy, cell death increased from 5-15% vs. 15-45% (3-fold) is not the same on tumor outcome or p-eIF2a levels. p-4E controls the ISR range.

16) Subsection “Mutant KRAS cooperates with p-4E/Myc to promote glutamine addiction and immune suppression”, Figure 6A. Total eIF4E levels are also elevated in Lim1215 cells with KRAS mutations, arguing against increased p-4E levels in this cell line. Quantification of immunoblot could help clarifying this.

Upon Gln- (by 24 hours), total GCN2, eIF4E or eIF2a did not change in any cell line examined. Myc reduction and ISR induction is more selective in mutant *KRAS* mutant cells (with higher basal levels). This is consistent with our model mutant *KRAS* cooperates with Myc via p-4E to promote Gln-dependent adaptation and addiction. See revised Figure 6D, Figure 6—figure supplement 1C, Figure 7F, and Results section.

17) Subsection “Mutant KRAS cooperates with p-4E/Myc to promote glutamine addiction and immune suppression”, Figure 6C,D. The apoptotic response in HCT116 wt cells is much lower than in previous assays in HCT116 (26% in Figure 6B vs. over 50% in Figure 5) and no apoptosis induction at all in Lim1215 wt cells upon GLN deprivation (also no or only weak induction of ISR, and no cl. Casp 3, Figure 6D). This raises the question to which extent this phenotype/mechanism can be generalized to CRC or whether the mutational landscape of different CRC cells affect the response. The number of analysed cell lines could be extended or a more physiological, genetically defined system with specific mutations such as intestinal organoids could be used.

Figure 5C. 4E WT *vs.* 4E KI HCT 116 cells (mutant *KRAS*). Figure 6C, WT and mutant refer to *KRAS (*not *4E S209),* the death was 50-60% in *G13D* cells. Therefore, apoptosis was reduced in 4EKI or WT *KRAS* HCT 116 cells similarly.

Between Figures 6 and 7, we have used over 10 different CRC lines, supporting Gln addiction is correlated with higher basal p-4E/p-4EBP1, Myc, ISR and transcriptional heterogeneity. Pharmacological Mnki reduced the growth of CRC organoids, p-4E and Myc levels. Gln is not used in organoid culture, and Myc-driven metabolic/Gln addiction can be compensated by alternative fuels. See new Figure 2I-J, revised Results and Discussion section.

18) The authors state “The sensitivity was associated with elevated apoptosis and p-GCN2/ISR response, and reduction in Myc but p-ERK (Figure 6C-E, Figure 6—figure supplement 1A-C)." MYC levels are not shown in any of the given figures. Why should MYC levels be reduced in KRASmut cells upon GLN deprivation if they are more sensitive to deprivation and if this is mediated via MYC induction?

Gln- induced more cell death and Myc reduction in HCT 116 cells (vs 4EKI or WT *KRAS*), Lim 1215 (mutant *KRAS* vs WT), and RKO, HT29 (vs. DLD1 and SW480). As depicted in revised model (Figure 7F), higher basal Myc and p-4E/p-4EBP1 leads to oncogenic addiction, i.e. the dependence on Myc and Gln for adaptation and growth. Their loss (acute) leads to increased stress (ISR) and cell death.

19) The authors state “These data establish that p-4E/Myc cooperates with mutant KRAS to promote glutamine addiction with elevated ISR and immune suppression". Are transcriptome analyses in Figure 6F-J performed in KRASwt cells? This is not completely obvious from the description. If yes, then the claim that cooperation with KRAS promotes immune suppression is not reflected by the data. Global gene expression changes are then only shown for KRASwt cells, same analyses need to be performed in KRASmut cells (-/+GLN, wt/4EKI).

Our focus is to understand the role of p-4E in shapin*g* the response of mutant *KRAS* to metabolic stress. See revised Results section.

20) Figure 7B. Total eIF4E and 4EBP levels are also elevated in RKO and HT29 cells, arguing against increased p-4E/p-4EBP levels. Quantification of immunoblot would help clarifying this.

We agree. However, total eIF4E and 4E-BP1 (eIF2a or GCN2) in these cells were unaffected by Gln-. SW480 cells have eIF4E but no detectable 4EBP1, supporting the heterogeneity of CRC, compared to more uniform p-4E in mouse polyps or human adenomas. This is also the very reason for us to examine these markers, ISR levels, and transcriptional response in a panel of CRCs including isogenic pairs.

21) The authors state “Together, our data support that p-4E controls cell death upon metabolic stress". This conclusion is only partially reflected by the data. I appreciate that HCT116 cells show reduced p-4E levels upon metabolic stress (GLN deprivation) associated with enhanced stress signaling (Figure 5). Instead, neither HT29 nor RKO cells that are sensitive to GLN deprivation with enhanced stress signaling, show changes in p-4E levels (Figure 7B). Therefore, it is questionable whether the proposed mechanism, p-4E regulating the response to metabolic stress, is a general feature of CRC. On the same line, a clear mechanism of the response to GLN deprivation in CRC cells has been described previously, that is associated with reduced nucleotide and MYC levels and, subsequently, slowed elongation of RNA Pol II and cell cycle arrest instead of apoptosis. This is not at all discussed in the manuscript.

See above and 11) too. This model helps explain that Myc-driven metabolic addiction likely goes beyond Gln to support increased biosynthesis. Metabolic stress (Gln-) leads to Myc loss and hyperactivation of ISR, with increased cell death and other oncogenic features. Therefore, the levels and duration of ISR dictates transcriptional response, TME interaction (i.e., immune signals) and outcome. See revised Figure 7F and Discussion section (2017 Embo paper and more papers cited).

22) The depicted model in Figure 7F is not comprehensible.

See 18, 20 and 21 too. We simplified the model to show how mutant *KRA*S (and other drivers) increases p-4E/p-4EBP1 and Myc/ISR and Gln-dependent growth. Gln deprivation reduced Myc and elevated ISR and cell death with other outcomes (transcriptional heterogeneity including immune suppression). Revised Figure 7F.